# A novel class of inferior colliculus principal neurons labeled in vasoactive intestinal peptide-Cre mice

David Goyer[1], Marina A Silveira[1], Alexander P George[1], Nichole L Beebe[2], Ryan M Edelbrock[2], Peter T Malinski[1], Brett R Schofield[2], Michael T Roberts[1]*

[1]Kresge Hearing Research Institute, Department of Otolaryngology – Head and Neck Surgery, University of Michigan, Ann Arbor, United States; [2]Department of Anatomy and Neurobiology, Northeast Ohio Medical University, Rootstown, United States

**Abstract** Located in the midbrain, the inferior colliculus (IC) is the hub of the central auditory system. Although the IC plays important roles in speech processing, sound localization, and other auditory computations, the organization of the IC microcircuitry remains largely unknown. Using a multifaceted approach in mice, we have identified vasoactive intestinal peptide (VIP) neurons as a novel class of IC principal neurons. VIP neurons are glutamatergic stellate cells with sustained firing patterns. Their extensive axons project to long-range targets including the auditory thalamus, auditory brainstem, superior colliculus, and periaqueductal gray. Using optogenetic circuit mapping, we found that VIP neurons integrate input from the contralateral IC and the dorsal cochlear nucleus. The dorsal cochlear nucleus also drove feedforward inhibition to VIP neurons, indicating that inhibitory circuits within the IC shape the temporal integration of ascending inputs. Thus, VIP neurons are well-positioned to influence auditory computations in a number of brain regions.
DOI: https://doi.org/10.7554/eLife.43770.001

*For correspondence:
microb@umich.edu

**Competing interests:** The authors declare that no competing interests exist.

## Introduction

The inferior colliculus (IC) is the hub of the central auditory pathway. Nearly all ascending output from the lower auditory brainstem and a large descending projection from the auditory cortex converge in the IC (*Adams, 1979*; *Glendenning and Masterton, 1983*; *Oliver, 1987*; *Oliver, 1984*; *Winer et al., 1998*). In turn, the IC provides the main auditory input to the thalamocortical system (*Calford and Aitkin, 1983*). Neurons in the IC exhibit selective responses to the spectral and temporal content of sounds and perform computations important for sound localization and the identification of speech and other communication sounds (*Felix et al., 2018*; *Winer and Schreiner, 2005*). Despite these critical functions, we have limited knowledge about the organization and function of neural circuits in the IC. This is because probing neural circuits requires the ability to identify and manipulate specific classes of neurons, but IC neurons have proven difficult to delineate into distinct classes.

Anatomical studies have shown that IC neurons have disc-shaped or stellate morphologies (*Malmierca et al., 1993*; *Meininger et al., 1986*; *Oliver and Morest, 1984*). Disc-shaped neurons maintain their dendritic arbors within isofrequency lamina and make up the majority of neurons in the tonotopically organized central nucleus of the IC (ICc). Stellate neurons in the ICc extend their dendritic arbors across laminae and are therefore thought to integrate information across sound frequencies (*Oliver et al., 1991*). Both disc-shaped and stellate cells can be glutamatergic or GABAergic, an indication that each of these morphological groups consists of at least two neuron types

**eLife digest** Our brains help us make sense of our surroundings. Our sense organs, for example the ears, receive signals from the environment, which are passed on to specialized parts of the brain, called nuclei. There, neurons process the information and send impulses to other areas in the brain, which eventually help us decide how to respond.

A nucleus called the inferior colliculus is one of the major hearing centres in the mammalian brain. In humans, it is vital for recognizing speech and pinpointing the location of sounds. It contains several different types of neurons, and links with many other areas of the brain.

While the inferior colliculus is crucial for our sense of hearing, little is known about the specific properties of the neurons within it. In particular, these cells are difficult to divide into well-defined groups. Even less is understood about the precise nature of the connections between these neurons, which likely underpin the computational power of the inferior colliculus. Goyer et al. therefore set out to identify a specific class of neurons in this region and map the circuits they formed.

Experiments were conducted on genetically modified mice whose neurons were only 'glowing' if they had a gene called *VIP* switched on. Detailed examination of the shape of the 'VIP cells', as well as their chemical and electrical properties, confirmed that they were indeed a distinct class of neurons.

Another set of experiments relied on a method that uses light to control the activity of brain cells. This showed that VIP cells received signals both from other neurons in the inferior colliculus, and from another hearing center in the brain. In turn, VIP cells sent signals over long distances to many other parts of the brain that handle sound signals. This suggests that VIP neurons have a wide-ranging influence on our brains' ability to process sound.

The work by Goyer et al. has, for the first time, reliably identified specific circuits in a brain region essential for our sense of hearing. By knowing more about how the brain's hearing centers are connected to each other, it may become possible to understand their roles in hearing loss. In this effort, the inferior colliculus may become a target for treatments for patients with hearing difficulties.

DOI: https://doi.org/10.7554/eLife.43770.002

(*Oliver et al., 1994*). Based on soma size and extracellular markers, IC GABAergic neurons have been divided into four classes (*Beebe et al., 2016*). Among these, 'large GABAergic' neurons are the one consistently identified neuron type in the IC (*Geis and Borst, 2013*; *Ito et al., 2015*; *Ito et al., 2009*; *Ito and Oliver, 2012*). However, there are currently no known molecular markers specific for large GABAergic neurons (*Schofield and Beebe, 2019*).

Defining IC neuron types based on physiology has also proven difficult. IC neurons exhibit diverse responses to tones, but a comprehensive study showed that these responses form a continuum and cannot be used on their own to define functionally significant groups of neurons (*Palmer et al., 2013*). In addition, disc-shaped neurons could not be divided into distinct groups by matching their morphology with their in vivo physiology (*Wallace et al., 2012*). Similarly, GABAergic and glutamatergic IC neurons exhibit overlapping and equally diverse responses to sounds (*Ono et al., 2017*). In vitro recordings have shown that IC neurons exhibit diverse firing patterns, but these firing patterns do not correlate with neuronal morphology or neurotransmitter phenotype (*Ono et al., 2005*; *Peruzzi et al., 2000*; *Reetz and Ehret, 1999*; *Sivaramakrishnan and Oliver, 2001*).

In many brain regions, a multidimensional analysis that includes molecular markers has proven key to identifying neuron classes (*Ascoli et al., 2008*; *Tremblay et al., 2016*; *Zeng and Sanes, 2017*). Here, by combining molecular, morphological, and physiological analyses, we identify vasoactive intestinal peptide (VIP) neurons as a novel class of IC principal neurons. Our results show that VIP neurons are glutamatergic stellate neurons and represent approximately 20% of the stellate neurons in the ICc. VIP neurons are labeled in the VIP-IRES-Cre mouse line and are present in the major subdivisions of the IC, with a higher prevalence in caudal regions of the ICc, where most ascending input originates from monaural brainstem nuclei. Using viral tract tracing, we found that VIP neurons project to multiple auditory and non-auditory areas, demonstrating that a single neuron class can

participate in most of the major projection pathways out of the IC. Using Channelrhodopsin-assisted circuit mapping (CRACM), we found that VIP neurons integrate input from the contralateral IC and the auditory brainstem. Input from the auditory brainstem also drove local, feedforward inhibition onto VIP neurons. Thus, our data reveal a novel circuit motif that may control the temporal summation of ascending input to the IC. Together, these results represent a critical step toward determining how defined neural circuits in the IC support sound processing.

## Results

### The VIP-IRES-Cre mouse line labels neurons in multiple subdivisions of the IC

By crossing VIP-IRES-Cre mice with Ai14 reporter mice, we obtained mice in which VIP+ neurons expressed the fluorescent protein tdTomato. Distribution of VIP+ neurons throughout the brain matched the description provided by *Taniguchi et al. (2011)*. There were very few or no VIP-expressing neurons in auditory centers outside of the IC (including cochlear nucleus, superior olivary complex, nuclei of the lateral lemniscus, nucleus of the brachium of the IC and medial geniculate nucleus), with the exception of auditory cortex, where sparse neurons matching descriptions of VIP-expressing interneurons were labeled. The present report focuses on the IC, which contained many VIP-expressing neurons. *Figure 1* shows the distribution of VIP+ neurons (magenta) in transverse sections through the IC. Labeled neurons were present throughout most of the rostro-caudal extent of the IC, including in the ICc, ICd, and IClc, but were most numerous in the caudal regions. Labeled neurons were rare or absent in the IC rostral pole and intercollicular tegmentum.

### VIP neurons are glutamatergic and represent 3.5% of ICc neurons

Previous studies have shown that IC neurons are either glutamatergic or GABAergic (*Merchán et al., 2005*; *Oliver et al., 1994*). To investigate the neurotransmitter content of VIP neurons, we performed immunohistochemical staining against GAD67, a marker for GABAergic neurons, in brain slices from three VIP-IRES-Cre x Ai14 animals, aged P58 (*Figure 2A*). We then counted tdTomato+ neurons and GAD67-labeled cell bodies in one caudal and one rostral IC slice per animal. Because there were no regional differences in GAD67 staining among VIP neurons, neurons located in the ICc, ICd, and IClc were combined for this analysis. Across 793 tdTomato+ neurons, only 10 neurons co-labeled with GAD67 (1.3%, *Table 1*). We suspect the 10 tdTomato+ neurons that stained for GAD67 were non-specifically labeled or represent rare cases of tdTomato expression in non-VIP

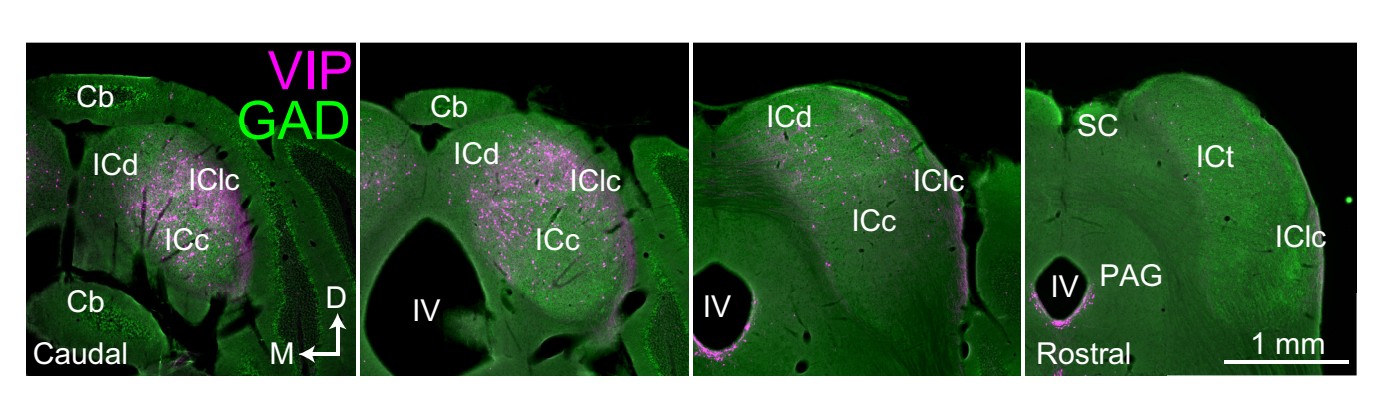

**Figure 1.** VIP neurons are distributed across multiple subdivisions of the IC. Photomicrographs of transverse sections through the IC ranging from caudal (left-most) to rostral (right-most). VIP-expressing cells (labeled with tdTomato) are shown in magenta, and GAD67 staining is shown in green to show the border of the IC. VIP-expressing cells are present in multiple subdivisions of the IC, but are most prominent in caudal and dorsal parts of the IC. Scale = 1 mm. Cb (cerebellum), ICc, ICd, IClc (central nucleus, dorsal cortex and lateral cortex of the inferior colliculus), ICt (intercollicular tegmentum), IV (fourth ventricle), PAG (periaqueductal gray).
DOI: https://doi.org/10.7554/eLife.43770.003

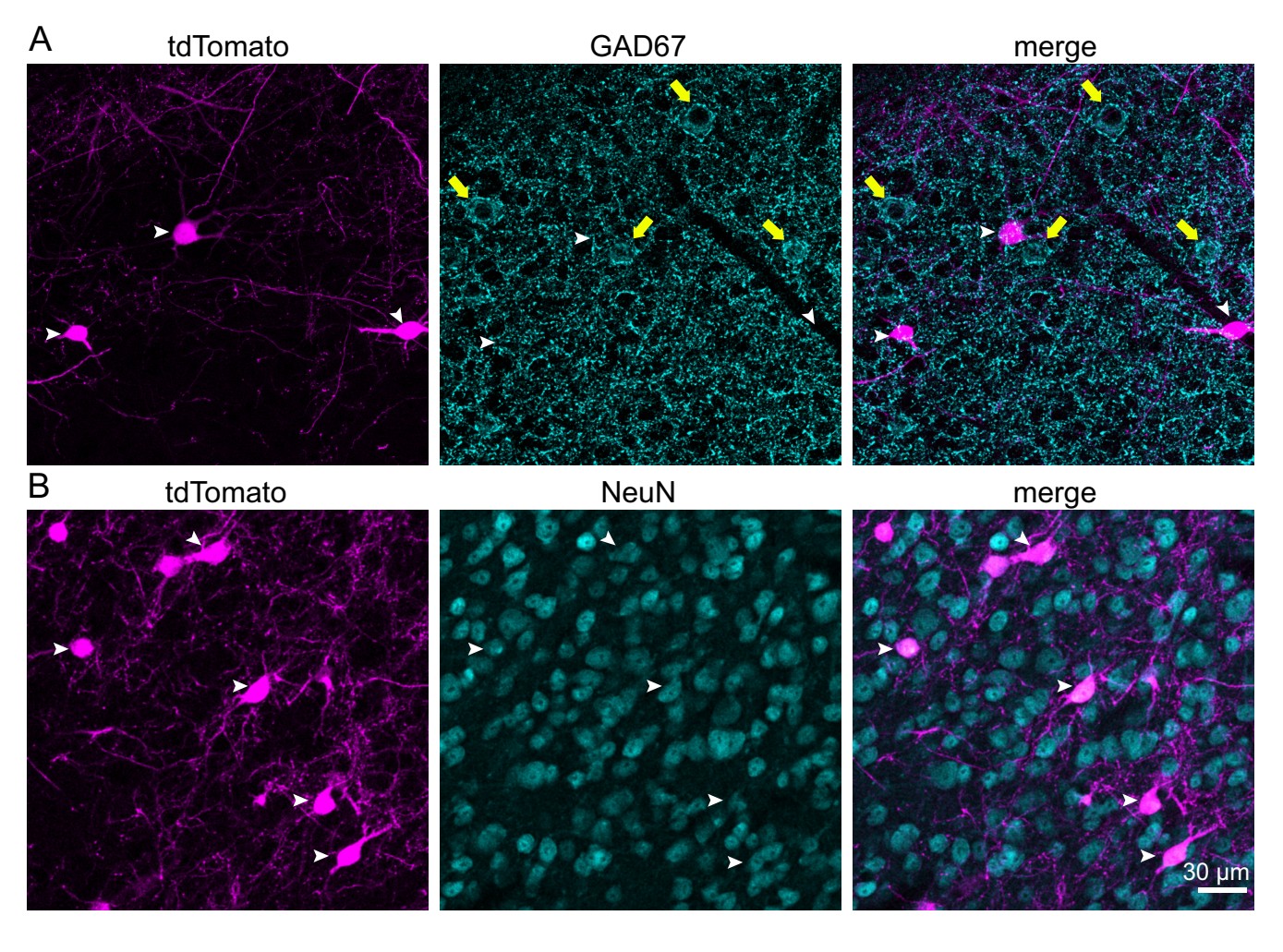

**Figure 2.** VIP neurons are glutamatergic and represent 3.5% of neurons in the ICc. (**A**) Confocal z-stack projections showing IC VIP neurons (magenta, left), GAD67 staining (cyan, middle), and an overlay (right). White arrowheads mark VIP neurons, yellow arrows GABAergic cell bodies. There was virtually no overlap between VIP neurons and GABAergic neurons (right). (**B**) Confocal z-stack projections showing VIP neurons (magenta, left), NeuN staining (cyan, middle), and an overlay (right). White arrowheads mark VIP neurons labeled by NeuN. Scale bar applies to A and B.
DOI: https://doi.org/10.7554/eLife.43770.005

neurons. These data suggest that VIP neurons are a subgroup of glutamatergic neurons in the ICc, ICd, and IClc.

To determine the percentage of neurons in the ICc and IC shell (ICd plus IClc) that are VIP neurons, we performed immunostaining with anti-NeuN, a neuron-selective antibody previously shown to label most or all neurons in the IC (*Beebe et al., 2016*; *Foster et al., 2014*; *Mellott et al., 2014*) (*Figure 2B*), and anti-bNOS, a marker that differentiates the ICc from the IC shell regions (*Coote and Rees, 2008*). Coronal IC sections from two VIP-IRES-Cres x Ai14 mice were stained with anti-NeuN and anti-bNOS. Three sections per mouse were analyzed: one caudal, one middle, and one rostral. To ensure unbiased counting of neurons, we applied the optical fractionator method, a design-based stereology approach (see Materials and methods; *West et al., 1991*). Accordingly, we used systematic random sampling to collect confocal image stacks with a 63x objective at evenly spaced intervals from each IC section. Each image stack was inspected to determine the boundaries of the slice, and guard zones were set at the top and bottom of the slice to delineate a central, 15 μm-thick section of the slice for subsequent analysis. Within this 15 μm-thick region, we separately marked NeuN$^+$ neurons and tdTomato$^+$ neurons, then overlaid the NeuN and tdTomato images to

**Table 1.** VIP neurons are glutamatergic.

Across three mice, an average of 1.3% of tdTomato$^+$ neurons were labeled with an antibody against GAD67.

| Animal | Slice # | # tdTomato$^+$ | # GAD67$^+$ | # co-labeled | % tdTomato$^+$co-labeled |
|---|---|---|---|---|---|
| P58 female, #1 | 1 (caudal) | 210 | 184 | 3 | 1.4 |
| | 2 (middle) | 172 | 65 | 2 | 1.2 |
| | Total | 382 | 249 | 5 | 1.3 |
| P58 male | 1 (caudal) | 151 | 152 | 2 | 1.3 |
| | 2 (middle) | 46 | 212 | 2 | 4.3 |
| | Total | 197 | 364 | 4 | 2.0 |
| P58 female, #2 | 1 (caudal) | 161 | 137 | 0 | 0.0 |
| | 2 (middle) | 53 | 187 | 1 | 1.9 |
| | Total | 214 | 324 | 1 | 0.5 |
| Grand total | | 793 | 937 | 10 | 1.3 |
| Average across three mice (mean ± SD) | | | | | 1.3 ± 0.8% |

DOI: https://doi.org/10.7554/eLife.43770.004

determine the number of double-marked cells. We also collected tile scan images of each IC section analyzed using a 20x objective and used bNOS staining to determine the border separating the ICc from the IC shell. The 63x z-stacks were aligned to these tile scans, and counted neurons were assigned to the ICc or IC shell.

The results of the stereological analysis revealed that VIP neurons represented a larger portion of neurons in the ICc (3.5 ± 1.0%) than in the IC shell (1.5 ± 0.2%; two-tailed t-test, $t_{(79)}$ = 2.86, p = 0.005; *Table 2*). In addition, the prevalence of VIP neurons was highest in caudal regions of the ICc and the IC shell and tended to decrease in a caudal to rostral gradient. This trend was significant when comparing the caudal ICc to the rostral ICc (one-way ANOVA, $F_{(2,37)}$ = 7.27, p = 0.002, Tukey's post hoc, p=0.001) and the caudal IC shell to the middle IC shell (one-way ANOVA, $F_{(2,38)}$ = 3.88, p = 0.03, Tukey's post hoc, p = 0.03). These results suggest that VIP neurons are not evenly distributed along the rostral-caudal extent of the isofrequency lamina of the ICc and, according to the functional domain hypothesis, may be more likely to receive input from ascending afferents that preferentially target the caudal ICc (*Cant and Benson, 2006*; *Loftus et al., 2010*; *Oliver et al., 1997*). The functional implications of this anatomical arrangement will be addressed in more detail in the Discussion. Overall, a combined count of VIP neurons from the ICc and IC shell showed that VIP neurons represent 2.3 ± 0.3% of the total population of neurons in the mouse IC (208 of 9304 neurons, n = 69 systematic random samples).

## VIP neurons exhibit sustained firing patterns and their intrinsic physiology varies along the tonotopic axis of the ICc

Next, we investigated the firing pattern and intrinsic physiology of VIP neurons by targeting whole cell patch clamp recordings to tdTomato$^+$ neurons in brain slices from VIP-IRES-Cre x Ai14 mice. For the majority of neurons, neuronal location relative to the IC subdivisions was determined post hoc, during retrieval of neuronal morphology (see below and Materials and methods). Recordings made from the ICc, ICd, and IClc were lumped together for this experiment because there were no clear differences in VIP neuron physiology across these subdivisions of the IC. VIP neurons had a resting membrane potential of −69.5 mV ± 4.4 mV (n = 216, corrected for liquid junction potential). In response to a current step protocol with hyperpolarizing and depolarizing current injections, VIP neurons showed minimal to no voltage sag to hyperpolarizing current steps and a sustained firing pattern of action potentials to depolarizing current steps (*Figure 3A$_1$, A$_2$*). Neurons were classified as sustained if their spike frequency adaptation ratio (SFA) was less than 2 (*Peruzzi et al., 2000*). The SFA ratio was calculated by dividing the last interspike interval by the first for a depolarizing current step that elicited ~10 spikes. 90.3% (214 of 237) of patched VIP neurons exhibited a sustained firing pattern, 8.4% (20 of 237) showed an adapting firing pattern (SFA ratio >= 2), and 1.3% (3 of 237) of VIP neurons had a transient firing pattern (firing stopped before the end of the current step).

**Table 2.** VIP neurons represent 3.5% of ICc neurons, 1.5% of IC shell neurons, and are present at a higher density in the caudal ICc and IC shell.

Table shows results from stereological analysis of the percentage of neurons (NeuN$^+$) in the ICc and IC shell (ICd + IClc) that express tdTomato in VIP-IRES-Cre x Ai14 mice. Values indicate mean ± SEM, (#tdTomato$^+$ neurons / #NeuN$^+$ neurons), and number of systematic random samples analyzed from each slice.

**ICc**

| Coronal slice | P54 male 1 | P54 male 2 | Per slice plane | Grand average |
|---|---|---|---|---|
| Caudal | 3.1 ± 0.9% (12/503) five samples | 8.4 ± 1.2% (26/338) four samples | 5.8 ± 2.7% (38/841) | |
| Middle | 2.4 ± 0.8% (20/741) eight samples | 3.9 ± 0.9% (44/1173) eight samples | 3.2 ± 0.7% (64/1914) | |
| Rostral | 1.9 ± 0.6% (21/929) eight samples | 1.2 ± 0.4% (12/1024) seven samples | 1.5 ± 0.4% (33/1953) | |
| Per mouse | 2.5 ± 0.3% (53/2173) | 4.5 ± 2.1% (82/2535) | | 3.5 ± 1.0% (135/4708) |

**IC shell**

| Coronal slice | P54 male 1 | P54 male 2 | Per slice plane | Grand average |
|---|---|---|---|---|
| Caudal | 2.9 ± 0.8% (35/1092) 10 samples | 1.9 ± 0.8% (9/615) five samples | 2.4 ± 0.5% (44/1707) | |
| Middle | 0.9 ± 0.6% (4/534) six samples | 0.9 ± 0.6% (10/944) eight samples | 0.9 ± 0.0% (14/1478) | |
| Rostral | 1.2 ± 0.4% (10/842) eight samples | 1.2 ± 0.6% (5/569) four samples | 1.2 ± 0.0% (15/1411) | |
| Per mouse | 1.6 ± 0.6% (49/2468) | 1.3 ± 0.3% (24/2128) | | 1.5 ± 0.2% (73/4596) |

DOI: https://doi.org/10.7554/eLife.43770.006

The following source data is available for Table 2:

Source data 1. Percentages of VIP neurons in ICc and IC shell.
DOI: https://doi.org/10.7554/eLife.43770.007

To compare the physiology of VIP neurons to that of the general population of neurons in the IC, we patched neurons in IC slices of C57BL/6J mice in a random, non-targeted approach as a control. Neurons patched with the non-targeted (NT) approach showed a higher diversity in firing patterns, with a higher proportion of transient neurons (21.8%, 12 out of 55) and adapting neurons (16.4%, 9 out of 55) when compared to VIP neurons. Sustained firing neurons were the most prevalent group in NT recordings (58.2%, 32 out of 55). Additionally, 3.6% (2 of 55) of randomly patched IC neurons fired only one spike at the onset of the depolarizing current step. This firing pattern was never observed in VIP neurons. The intrinsic physiology of VIP neurons also differed significantly from the general population of IC neurons (see *Figure 3B*). VIP neurons on average had a higher peak input resistance ($R_{pk}$) than non-targeted neurons (mean ± SD: VIP 242.1 ± 139.4 MΩ vs NT 191.1 ± 161.4 MΩ, p = 0.0003, Wilcoxon rank sum test), a higher steady-state input resistance ($R_{ss}$) (mean ± SD: VIP 239.7 ± 170.7 MΩ vs NT 153.4 ± 153.2, p = 0.0001, Wilcoxon rank sum test), a slower membrane time constant (mean ± SD: VIP 15.0 ± 8.8 ms vs NT 9.7 ± 7.6 MΩ, p = 6.8*10$^{-7}$, Wilcoxon rank sum test), lower rheobase values (mean ± SD: VIP 67.8 ± 96.2 pA vs NT 120.0 ± 100.9 pA, p = 0.013, Wilcoxon rank sum test), and a much less pronounced voltage sag (mean ± SD: VIP 0.87 ± 0.16 vs NT 0.75 ± 0.21, p = 0.0003, Wilcoxon rank sum test). Most striking, the SFA ratio of VIP neurons was

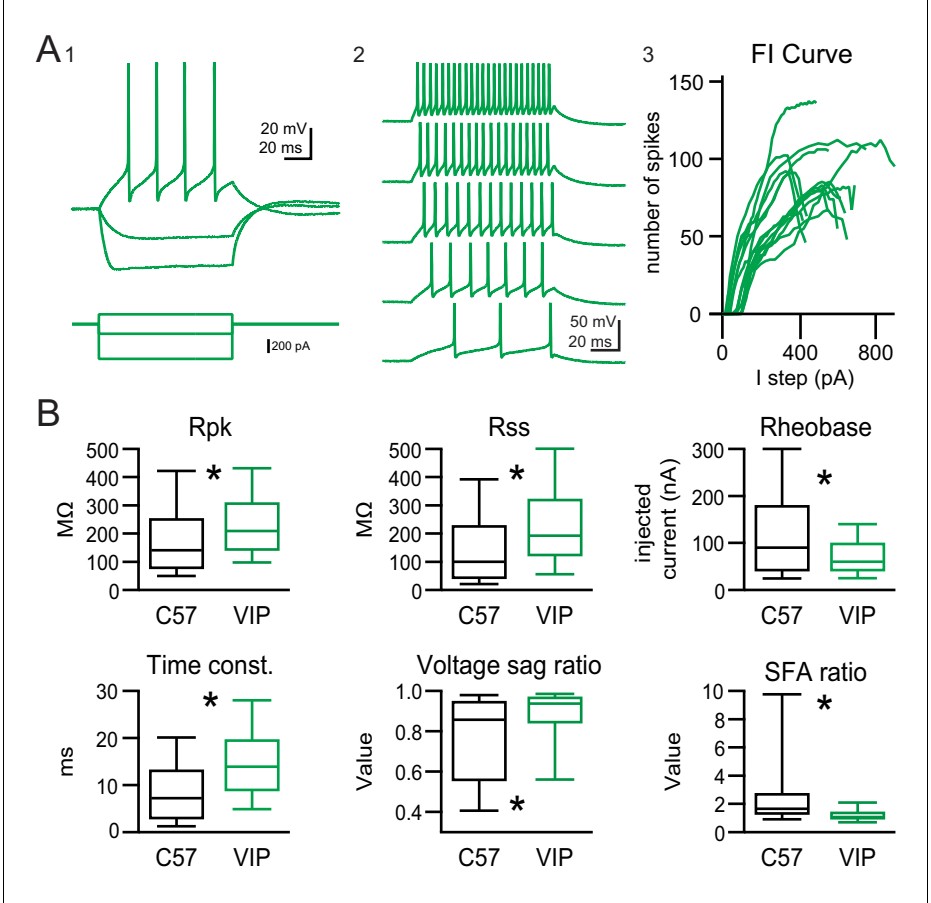

**Figure 3.** VIP neurons have sustained firing patterns and moderate membrane properties. (A) VIP neurons exhibited a regular, sustained firing pattern in response to depolarizing current steps, while hyperpolarizing current steps elicited minimal voltage sag (A₁). As the amplitude of depolarizing current steps was increased, VIP neurons increased their firing rate while keeping their sustained firing pattern (A₂). Example firing versus input (FI) curves from 15 VIP neurons show that firing rate increased in a mostly linear fashion over a broad range of current step amplitudes (A₃). (B) Intrinsic physiology of VIP neurons is statistically different from the general population of IC neurons for all parameters tested. On average, VIP neurons had a significantly higher peak input resistance ($R_{pk}$) and steady-state input resistance ($R_{ss}$), a lower rheobase, a longer membrane time constant, a smaller and less variable voltage sag ($I_h$) at −91 mV, and a markedly small and highly invariable spike frequency adaptation ratio (SFA). Boxplots show median, 25th and 75th percentile (box), and 9th and 91th percentile (whiskers).

DOI: https://doi.org/10.7554/eLife.43770.008

The following source data is available for figure 3:

**Source data 1.** Intrinsic physiology of VIP neurons and from non-targeted recordings in mouse IC.

DOI: https://doi.org/10.7554/eLife.43770.009

tightly clustered at 1.47 ± 1.62, whereas SFA of NT neurons showed a significantly higher value and spread (3.62 ± 5.43, mean ± SD, p = 2.07*10⁻⁷).

Although statistically different from the general neuronal population in the IC and showing a tightly clustered SFA ratio, the intrinsic physiology of VIP neurons still showed some level of variability. In the lower auditory brainstem, it has been found that the intrinsic physiology of some neurons varies along the tonotopic axis (*Baumann et al., 2013*; *Hassfurth et al., 2009*). We therefore hypothesized that the intrinsic physiology of VIP neurons in the ICc varied along the tonotopic axis of the ICc. During patch clamp experiments, VIP neurons were passively filled with biocytin via the internal solution. Slices were fixed and stained post hoc with a streptavidin-Alexa Fluor conjugate (see Materials and methods). We then used confocal imaging to map the location of the recorded neurons relative to a two-dimensional (medial-lateral and dorsal-ventral) coordinate system

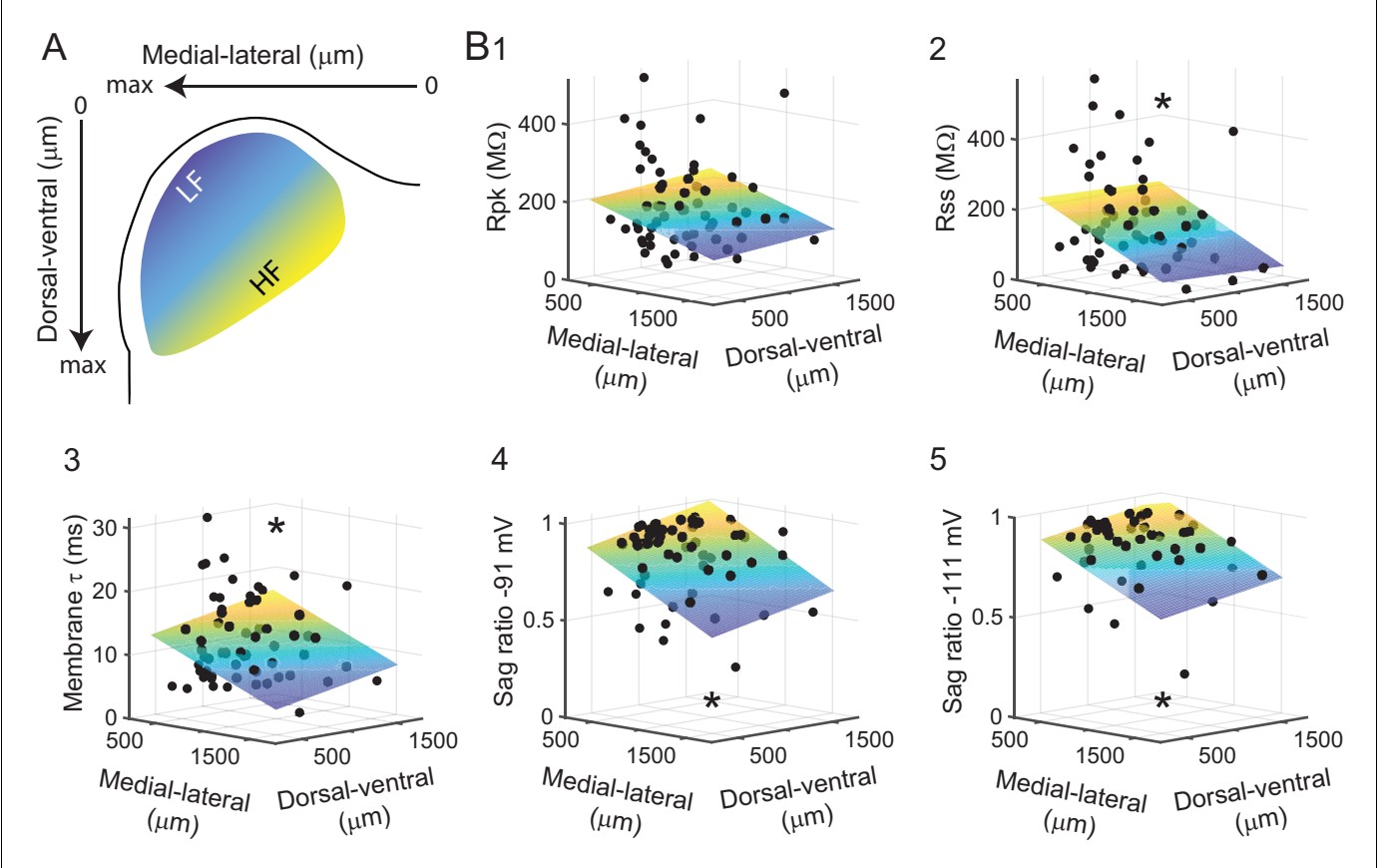

**Figure 4.** Intrinsic physiology of VIP neurons in the ICc varies along the tonotopic axis. (**A**) A 2D coordinate system was fit to every IC slice a VIP neuron was recorded from. The medial-lateral axis runs from the midline (zero) to the lateral edge (max) of the slice, the dorsal-ventral axis from the dorsal edge of the slice (zero) to the ventral border of the IC (max). For illustrative purposes, the approximate position along the tonotopic axis of the ICc is color-coded from blue (low frequency) to yellow (high frequency). (**B**) Correlation of measured intrinsic parameters with recording location. Black dots represent physiological parameters of individual VIP neurons (z-axis, left) mapped to their recording location (x- and y-axes, bottom). Planes show Levenberg-Marquardt least squares fits, color-coded from low z-axis values (blue) to high z-axis values (yellow). Asterisks indicate statistical significance of fit.

DOI: https://doi.org/10.7554/eLife.43770.010

The following source data is available for figure 4:

**Source data 1.** Intrinsic physiology of VIP neurons matched to their location in the ICc.

DOI: https://doi.org/10.7554/eLife.43770.011

superimposed on the left IC (n = 61 neurons; *Figure 4A*). Correlations between intrinsic physiology and location in the ICc were tested by fitting a plane to scatter plots of intrinsic parameters versus medial-lateral and dorsal-ventral coordinates (*Figure 4B*). Because the ICd and IClc are not tonotopically organized, only neurons located in the ICc were included in this analysis.

We found that variability in the intrinsic physiology of VIP neurons was at least partially correlated to their location within the coronal plane of the ICc. This was particularly true for the voltage sag ratios, which measure hyperpolarization-activated cation current ($I_h$). Approximately one quarter of the variability in sag ratios was explained by location in the ICc (sag ratio at −91 mV: R = 0.536, $R^2_{adj}$ = 0.262, p = 1.24×10$^{-05}$, n = 60, *Figure 4B₄*; sag ratio at −111 mV: R = 0.516, $R^2_{adj}$ = 0.233, p = 0.0002, n = 47, *Figure 4B₅*). A significant but smaller portion of the variability in membrane time constant was explained by location in the ICc (τ: R = 0.343, $R^2_{adj}$ = 0.088, p = 0.007, n = 61; *Figure 4B₃*). There was also a significant relationship between the steady-state input resistance of VIP neurons and location in the ICc and a trend toward a relationship between peak input resistance and location ($R_{ss}$: R = 0.328, $R^2_{adj}$ = 0.076, p = 0.011, n = 60; $R_{pk}$: R = 0.227, $R^2_{adj}$ = 0.018, p = 0.084, n = 60; *Figures 4B1* and *2*). The tonotopic axis of the ICc runs along a dorsolateral (low

frequency) to ventromedial (high frequency) axis (*Malmierca et al., 2008*; *Portfors et al., 2011*; *Stiebler and Ehret, 1985*; *Willott and Urban, 1978*). For each of the above intrinsic parameters, values tended to be lower, indicating faster membrane properties, at more dorsolateral locations and higher, indicating slower membrane properties, at more ventromedial locations. Combined, these results suggest that variability in the intrinsic physiology of VIP neurons is at least in part due to their localization along the tonotopic axis of the ICc and that the membrane properties of VIP neurons tend to be faster in lower frequency regions of the ICc.

## VIP neurons have stellate morphology and dendritic spines

The streptavidin staining of biocytin-filled VIP neurons allowed for a detailed analysis of morphology. In total, we recovered the morphology of 55% of patched VIP neurons (n = 100 of 183). Nearly all (81/86 = 94.2%) VIP neurons had spiny dendrites (*Figure 5A,B* insets). This contrasts sharply with the 28% (12 of 43 neurons) of neurons that had spiny dendrites in non-targeted recordings from C57BL/6J mice (*Figure 5C*), suggesting that VIP neurons represent a subset of IC cells. Nonetheless, dendritic spines can be present on a variety of cell types, including stellate and disc-shaped cells (*Herrera and Correa, 1988*; *Paloff et al., 1992*; *Willard and Ryugo, 1983*). On average, VIP neurons had five primary dendrites (mean ± SD: 4.77 ± 1.38) that spread out in all directions from the soma, consistent with a stellate morphology. This is unsurprising in the ICd and IClc, where stellate morphology predominates, but warranted further analysis in the ICc, where stellate cells can have oriented dendritic trees but are outnumbered by the more highly oriented disc-shaped cells (*Malmierca et al., 1993*; *Oliver and Morest, 1984*; *Willard and Ryugo, 1983*). There were no obvious differences between the morphology of VIP neurons in the ICc versus the ICd or IClc, but this question warrants more detailed analysis in a future study.

*Figure 5D* shows the variability in the morphology of VIP neurons located in the ICc. Neurons are displayed as they would appear in a coronal slice of the left IC viewed from a caudal perspective. Oliver and colleagues distinguished disc-shaped from stellate neurons by calculating the length to width ratio of the dendritic arbor: neurons with a ratio <3 were stellate and those with a ratio >= 3 were disc-shaped (*Oliver et al., 1991*). Applying this classification to our sample, 93% of VIP neurons in the ICc (39 of 42 neurons) had a length to width ratio <3, therefore being classified as stellate (*Figure 5H*). Only three VIP neurons from the ICc had a length to width ratio >3. These results demonstrate that the dendritic arbors of VIP neurons are not as highly oriented as disc-shaped neurons, again consistent with the hypothesis that VIP neurons are a class of stellate neurons.

Although less oriented than disc-shaped cells, the VIP dendrites tended to show some orientation that could influence the range of frequencies that converge on the cell. To measure the orientation of ICc VIP neurons in relation to the isofrequency laminae, which in mouse run in a ~45° angle through the ICc (*Stiebler and Ehret, 1985*), we identified the longest and second longest axis of each neuron through principal component analysis and plotted the orientation of these axes on a standardized model of the IC (*Figure 5E*). No preferred orientation was apparent (*Figure 5E*, *combined*). Only 17% of VIP neurons (7 of 42) had their longest axis oriented within ± 15° of the 45° laminar plane, indicating that the dendritic arbors of most VIP neurons (83%, 35 of 42) may be positioned to cross one or more isofrequency laminae in the ICc (*Figure 5F*). To quantify this, we calculated the length the dendritic arbor extended perpendicular to a 45° laminar plane. The dendritic arbors of 83% of ICc VIP neurons (35 of 42) spread more than 100 μm perpendicular to the laminar plane, and more than 36% (15 of 42) spread more than 200 μm across the laminar plane (*Figure 5G*). Previous work in rats has shown that the isofrequency laminae have a center-to-center distance that ranges from 90 to 150 μm, while neurons contained within a lamina had a thickness ranging from 30 to 70 μm (*Malmierca et al., 1993*). If we assume that laminar dimensions in mouse are no thicker than those in rats, our results indicate that the dendritic fields of VIP neurons usually extend beyond at least one isofrequency lamina, consistent with the conclusion that VIP neurons in the ICc are a class of stellate neurons. Previous estimates indicate that approximately 15 – 20% of ICc neurons are stellate (*Oliver et al., 1991*; *Oliver and Morest, 1984*). Since our stereological analysis showed that VIP neurons represent 3.5% of ICc neurons, our results indicate that VIP neurons account for ~18 – 23% of ICc stellate neurons.

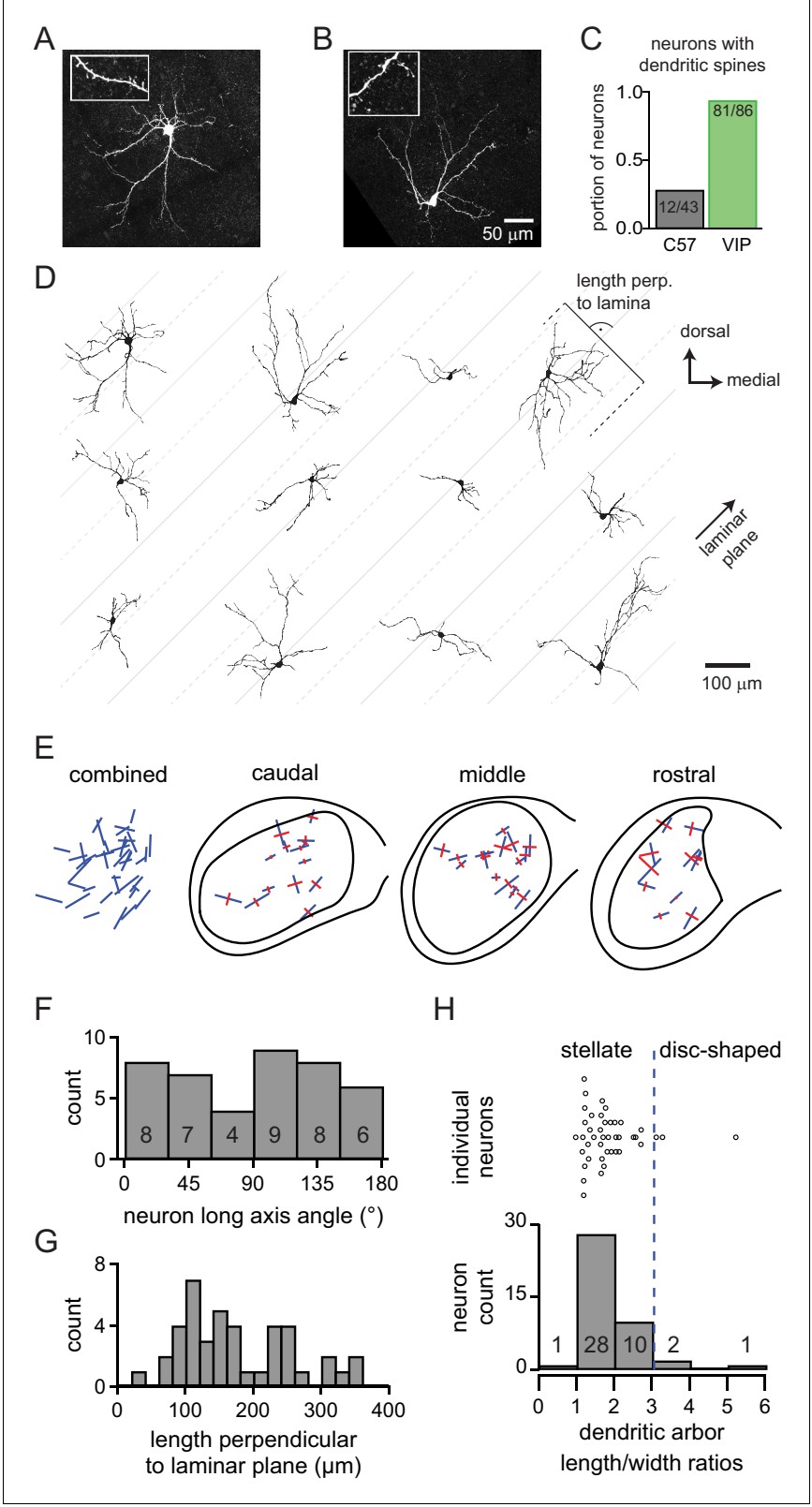

**Figure 5.** VIP neurons in the ICc are a class of stellate cells and most VIP neurons have dendritic spines. (A, B) Maximum-intensity projections of confocal z-stacks showing streptavidin-Alexa Fluor-stained VIP neurons from the ICc. Insets: enlarged views of dendritic segments show dendritic spines. (C) 94% of VIP neurons across all IC subdivisions had spiny dendrites vs 28% of neurons from non-targeted recordings in C57BL/6J animals. (D)
*Figure 5 continued on next page*

*Figure 5 continued*

Representative reconstructions of the morphology of 12 VIP neurons from the ICc. Neurons are oriented as if in the left ICc. Gray lines were drawn at a 45° angle to illustrate the general orientation of the laminae. Solid gray lines are spaced 200 µm apart, dashed lines and solid lines are spaced 100 µm apart. (E) Orientation of the dendritic fields of VIP neurons from the ICc. Combined: Orientation of all reconstructed VIP neurons from the ICc (n = 42). Blue lines denote the orientation of the longest axis (first principal direction) found for each neuron using 2D PCA. Caudal, middle, rostral: Orientation of dendritic fields separated according to position along the rostro-caudal axis of the ICc. Blue lines show longest axis, perpendicular red lines show second longest axis (second principal direction) of each neuron as defined by 2D PCA. (F) Angular orientation of the long axis for every reconstructed VIP neuron within the ICc. Angles indicate counter-clockwise rotation relative to the medial-lateral (horizontal) axis. (G) Spread of the dendritic arbors of ICc VIP neurons measured perpendicular to a predicted 45° isofrequency plane. The dendrites of 83% of VIP neurons extended more than 100 µm across the laminar plane. (H) Dendritic arbor length to width ratio for all reconstructed VIP neurons from the ICc (n = 42). 93% of VIP neurons had a length to width ratio <3, indicating that they are stellate cells. The orientation of length and width axes was determined using 3D PCA.

DOI: https://doi.org/10.7554/eLife.43770.012

The following source data is available for figure 5:

**Source data 1.** Morphometric analysis of VIP neurons in the ICc.
DOI: https://doi.org/10.7554/eLife.43770.013

## VIP neurons project to targets within and beyond the IC

Injections of an AAV encoding a Cre-dependent eGFP construct, AAV1.CAG.FLEX.eGFP.WPRE. bGH, led to eGFP expression in VIP+ IC cells. *Figure 6A* shows a representative deposit site, with eGFP-positive neurons (yellow) located among a population of VIP+ cells (magenta, labeled by cross-breeding the VIP-IRES-Cre mice with Ai14 reporter mice). Neurons that expressed eGFP routinely co-expressed tdTomato, confirming VIP expression by those neurons (*Figure 6B*). Many tdTomato+ neurons did not express the eGFP, despite their intermingling with many virally-labeled neurons. eGFP-labeled axons were prominent within the injected IC, where the labeled boutons were located in the neuropil or in close apposition to IC somas, suggesting extensive contributions to local circuits (*Figure 6C*). In addition, eGFP-labeled axons were present in several fiber tracts carrying projections from the IC, including the brachium of the IC, the commissure of the IC and the lateral lemniscus. Labeled axons and boutons were found in numerous auditory nuclei, including the contralateral IC, medial geniculate body and superior olivary complex (*Figure 6D–F*). Additional targets included the nucleus of the brachium of the IC, the periaqueductal gray, and the superior colliculus (not shown; details of termination patterns and terminal axon morphology will be described in a subsequent report). These data indicate that VIP+ IC neurons contribute to ascending, commissural and descending pathways from the IC.

## VIP neurons in the ICc receive excitatory and inhibitory synaptic input from the contralateral IC

In addition to axonal projection patterns, the sources of synaptic input to a neuron class are an important predictor of neuronal function. Anatomical studies have shown that the IC receives ascending, descending, and commissural input, but, with the exception of large GABAergic neurons (*Ito et al., 2015*; *Ito and Oliver, 2014*), it has rarely been possible to identify the sources and physiology of synaptic input to a specific class of neurons in the IC. This is largely because axons from multiple sources overlap in the IC, making it difficult to use electrical stimulation to selectively activate axons from specific presynaptic sources. In addition, electrical stimulation of commissural projections cannot differentiate between axons originating in the ipsilateral and contralateral IC. This is because electrical stimulation evokes both orthodromic and antidromic spikes, and, since most IC neurons have local axon collaterals (*Oliver et al., 1991*), antidromic spikes can lead to synaptic release from the local collaterals of neurons ipsilateral to the recording site. Furthermore, the commissure contains axons from sources other than IC cells, including cells in the auditory cortex, sagulum, nuclei of the lateral lemniscus and superior paraolivary nucleus (reviewed by *Saldaña and Merchán, 2005*); electrical stimulation in the commissure could also activate these axons. To overcome these obstacles, we turned to Channelrhodopsin-assisted circuit mapping (CRACM)

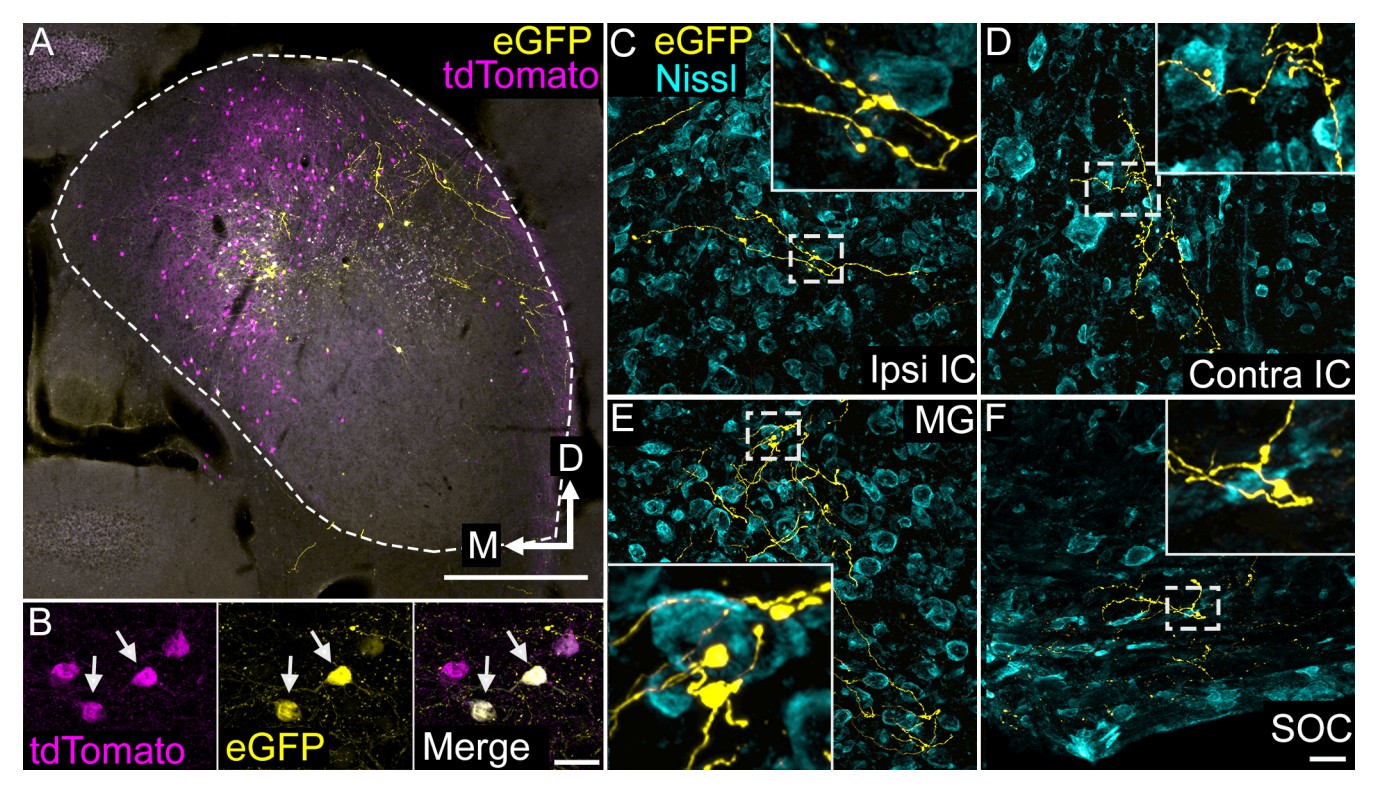

**Figure 6.** VIP neurons project to multiple local and long-range targets. (**A**) Photomicrograph of an AAV deposit site in the right IC. AAV-infected, VIP-expressing cells are labeled with eGFP (yellow), while all VIP-expressing cells are labeled with tdTomato (magenta). Cells expressing both fluorescent proteins appear white. Scale = 500 μm. (**B**) High magnification photomicrographs showing labeled cells in the AAV deposit site. The field shows four tdTomato-expressing cells (magenta), two of which (white arrows) were also AAV-infected and expressed eGFP (yellow). Scale = 20 μm. (**C–F**) High magnification photomicrographs showing eGFP-labeled collicular axons (yellow) terminating in the ipsilateral IC (**C**), the contralateral IC (**D**), the medial geniculate body (**E**), or the ventral nucleus of the trapezoid body in the superior olivary complex (**F**) after an AAV injection in the IC. The white dashed box in each image identifies an area enlarged in the inset to show details of labeled axons and boutons. A fluorescent Nissl stain (cyan) shows that boutons are located in close association with cell bodies as well as in the intervening neuropil. Scale = 20 μm.
DOI: https://doi.org/10.7554/eLife.43770.014

(*Petreanu et al., 2007*). With CRACM, it is possible to selectively activate a single population of pre-synaptic neurons by anatomically and/or molecularly restricting the expression of an optogenetic protein. In our experiments, we used stereotaxic coordinates and intracranial virus injections to anatomically restrict Chronos expression. Epi-fluorescence imaging confirmed that Chronos-GFP expression was always clearly limited to the right IC (*Figure 7B*) or the right DCN (*Figure 8A*).

Numerically, the contralateral IC provides the largest single source of input to the IC (*Moore, 1988*). We therefore first used CRACM to test whether VIP neurons in the ICc receive com-missural input. Using stereotaxic, intracranial virus injections with AAV1.Syn.Chronos-GFP.WPRE. bGH, we drove expression of GFP-tagged Chronos, a fast Channelrhodopsin variant (*Klapoetke et al., 2014*), in the right IC. Note that we did not attempt to limit Chronos expression to a particular subdivision of the right IC. Visual inspection of Chronos-GFP fluorescence suggested that most of the somata labeled in the right IC were located in the ICc, but labeled somata were often also present in the ICd and occasionally the IClc.

We then targeted recordings to VIP neurons in the contralateral (left) ICc (*Figure 7A*). In each experiment, we used GFP fluorescence to verify transfection of the right IC (*Figure 7B*) and restricted our recordings to VIP neurons in areas of the left ICc where GFP-labeled axons were visible. Because commissural projections are a mixture of glutamatergic and GABAergic projections (*González-Hernández et al., 1996*; *Hernández et al., 2006*; *Nakamoto et al., 2013*; *Saint Marie, 1996*), we used pharmacology to isolate EPSPs and IPSPs. Indeed, without adding receptor

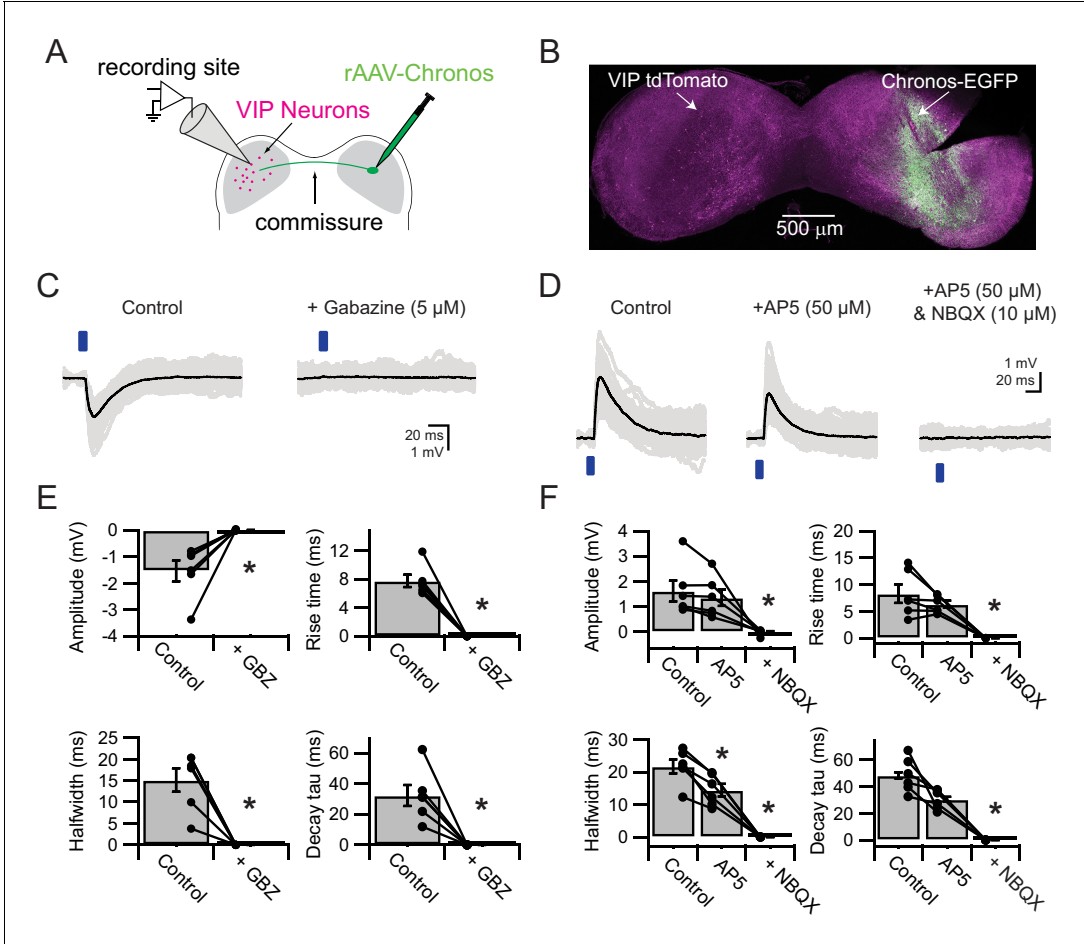

**Figure 7.** VIP neurons in the ICc receive excitatory and inhibitory synaptic input from the contralateral IC. (**A**) Experimental setup. An AAV encoding Chronos-GFP was injected into the right IC. Three weeks later, light-evoked postsynaptic potentials were recorded from VIP neurons in the left ICc. (**B**) Image of a coronal slice of the IC. Injection sites and Chronos expression were validated through Chronos-GFP fluorescence. (**C**) Optogenetically-evoked IPSPs recorded from VIP Neurons in the ICc contralateral to the AAV injection site. IPSPs were evoked by 2–5 ms blue light flashes (left), while EPSPs were blocked with NBQX and AP5. IPSPs were abolished by gabazine (right). (**D**) Optogenetically-evoked EPSPs recorded from VIP neurons in the ICc contralateral to the AAV injection site. EPSPs were evoked by 2–5 ms blue light flashes (left), while IPSPs were blocked with strychnine and gabazine. Wash-in of AP5 significantly reduced the halfwidth and decay time constant of light-evoked EPSPs (middle). Wash-in of NBQX abolished the remaining EPSP (right). (**E**) Population data showing amplitude and kinetics of optogenetically-evoked IPSPs. (**F**) Population data showing amplitude and kinetics of optogenetically-evoked EPSPs. The significant reduction of EPSP halfwidth by AP5 indicates that NMDA receptor activation prolonged EPSP duration.

DOI: https://doi.org/10.7554/eLife.43770.015

The following source data is available for figure 7:

**Source data 1.** EPSP and IPSP analysis of commissural inputs to VIP neurons.

DOI: https://doi.org/10.7554/eLife.43770.016

antagonists to the bath, postsynaptic potentials often were mixtures of IPSPs and EPSPs (data not shown). In the presence of AMPA and NMDA receptor antagonists (10 µM NBQX and 50 µM D-AP5, bath application), 2 – 5 ms flashes of blue light elicited IPSPs in 6 out of 12 neurons tested (*Figure 7C*, **left**). IPSPs were completely abolished by the GABA$_A$ receptor antagonist gabazine (5 µM, *Figure 7C*, right; n = 6; amplitude, p = 0.006; rise time, p = 0.003; halfwidth, p = 0.001; membrane time constant, p = 0.003, paired t-test). On average (n = 6), commissural IPSPs in ICc VIP neurons were small (−1.53 mV ± 0.96 mV) and had moderate 10 – 90% rise times (7.8 ms ± 2.1 ms), halfwidths (15.1 ms ± 6.8 ms) and decay time constants (32.4 ms ± 17.0 ms) (*Figure 7E*).

To investigate excitatory commissural inputs to ICc VIP neurons, recordings were carried out in the presence of GABA$_A$ and glycine receptor antagonists (5 µm gabazine and 1 µM strychnine, bath

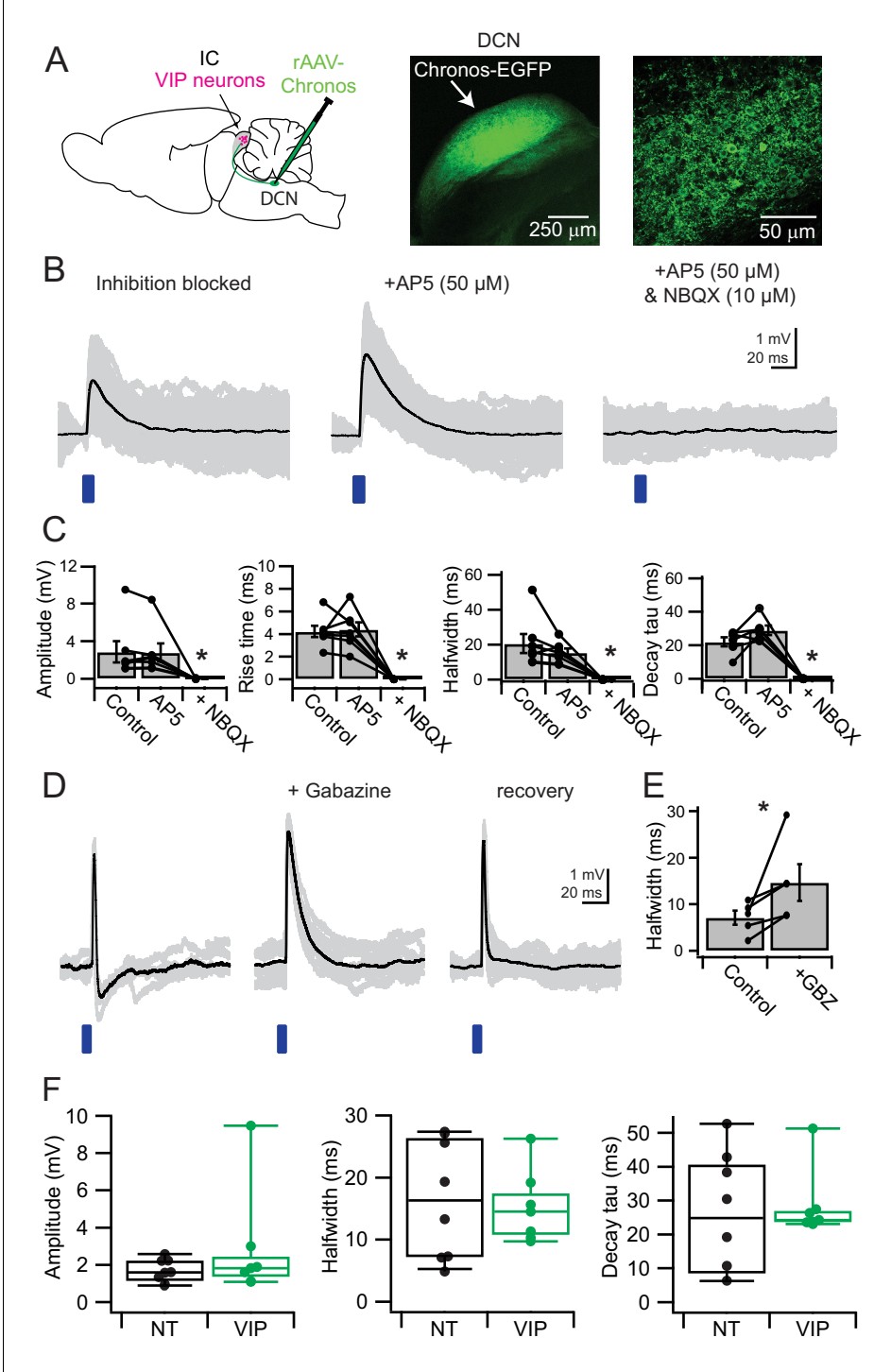

**Figure 8.** VIP neurons in the ICc receive direct synaptic input from the DCN and feedforward inhibition driven by DCN afferents. (**A**) Experimental setup. An AAV encoding Chronos-GFP was injected into the right DCN. For every experiment, the injection site and Chronos-GFP expression were confirmed through GFP fluorescence. Current clamp recordings were made from VIP neurons in the ICc contralateral to the injection site. (**B**) With inhibition blocked by gabazine and strychnine, 2–5 ms blue light flashes evoked EPSPs (left). AP5 did not significantly reduce EPSP halfwidth or decay time constant. Subsequent addition of NBQX abolished the EPSP. (**C**) Population data showing amplitude and kinetics for EPSPs elicited by activation of DCN synapses onto VIP neurons in the ICc. The absence of a significant effect of AP5 indicates that NMDA receptors did not make a significant contribution to EPSPs. (**D**) In several recordings made in the absence of inhibitory blockers, EPSP duration was limited through

*Figure 8 continued on next page*

*Figure 8 continued*

GABAergic feedforward inhibition (left; n = 5). In these instances, gabazine wash-in increased EPSP halfwidth to values similar to those in (B). (E) Population data for feedforward inhibition to VIP neurons. Washing in gabazine increased EPSP halfwidth in 5 out of 5 tested connections. (F) Population data comparing amplitude, halfwidth, and decay time constant of EPSPs from DCN inputs recorded in VIP neurons (VIP) and non-VIP neurons (NT, non-targeted recording). Halfwidth and decay time constants in VIP neurons showed a trend to cluster more tightly that in non-targeted recordings.

DOI: https://doi.org/10.7554/eLife.43770.017

The following source data is available for figure 8:

**Source data 1.** Analysis of EPSPs from DCN inputs in VIP neurons and non-targeted recordings.

DOI: https://doi.org/10.7554/eLife.43770.018

application, *Figure 7D*). We found that 2 – 5 ms flashes of blue light elicited EPSPs in 11 out of 27 neurons (*Figure 7D*, left). On average (n = 6), commissural EPSPs in VIP neurons were small (1.52 mV ± 1.08 mV) and had moderate 10 – 90% rise times (8.3 ms ± 4.3 ms), halfwidths (19.6 ms ± 7.6 ms) and decay time constants (43.5 ms ± 16.8 ms) (*Figure 7F*). Adding the NMDA receptor antagonist D-AP5 to the bath significantly reduced the halfwidth of EPSPs (14.3 ms ± 4.7 ms, p = 0.006) and revealed a trend toward reducing the rise time (6.3 ms ± 1.6 ms, p = 0.09) and decay time constant (30.6 ms ± 7.3 ms, p = 0.06) of EPSPs (ANOVA for repeated measurements with Tukey post-hoc test). The remainder of the EPSP was completely blocked by the AMPA receptor antagonist NBQX (*Figure 7F*). These results suggest that VIP neurons in the ICc receive excitatory commissural input and express both AMPA and NMDA receptors at excitatory commissural synapses. Interestingly, commissural input activated NMDA receptors even though there was 1 mM $Mg^{2+}$ in the bath and the somatic membrane potential was at or near the resting membrane potential throughout the recording. This may indicate that commissural synapses are located on the distal dendrites and/or the dendritic spines of VIP neurons.

Combined, the commissural CRACM experiments show that VIP neurons in the ICc receive excitatory and inhibitory synaptic input from the contralateral IC. Surprisingly, although GABAergic neurons make up <= 20% of commissural projections (*Hernández et al., 2006*; *Nakamoto et al., 2013*), we found a higher connection probability for inhibitory commissural projections (50%; 6 out of 12 recordings performed in the presence of NBQX and D-AP5), than for excitatory connections (41%; 11 out of 27 recordings performed in the presence of gabazine and strychnine).

Due to the kinetics of optogenetic proteins and the possibility of activating optogenetic proteins in synaptic terminals, optogenetically-evoked PSPs may have slower kinetics than PSPs evoked by electrical stimulation (*Jackman et al., 2014*; *Zhang and Oertner, 2007*). To test whether the kinetics of Chronos-evoked commissural PSPs were similar to those of electrically-evoked PSPs, we electrically stimulated the IC commissure while recording PSPs from VIP neurons in the contralateral IC. As indicated above, this raises the complication that electrically-evoked EPSPs were probably elicited both from the desired contralateral projections and from antidromic stimulation of ipsilateral axons. Compared to optogenetically-evoked EPSPs, electrically-evoked EPSPs (n = 6) had similar amplitudes and rise times, but had smaller halfwidths and trended toward faster decay time constants (amplitude: 2.4 mV ± 1.7 mV, p = 0.37; rise time: 6.28 ms ± 6.3 ms, p = 0.40; halfwidth: 10.3 ms ± 4.8 ms, p = 0.008*; decay time constant 28.4 ms ± 10.0 ms, p = 0.016, two tailed t-test, *critical p value with Bonferroni correction for multiple comparisons = 0.0125). Electrically-evoked IPSPs (n = 6) showed no significant difference in amplitude, halfwidth or decay time constant, but trended toward faster rise times (amplitude: −1.7 mV ± 1.3 mV, p = 0.78; rise time: 4.2 ms ± 2.3 ms, p = 0.025; halfwidth: 8.8 ms ± 7.3 ms, p = 0.16; decay time constant 36.8 ms ± 17.3 ms, p = 0.67, two tailed t-test, critical p value with Bonferroni correction for multiple comparisons = 0.0125). It should be noted that to position the stimulus electrode in the IC commissure contralateral to the recording site, these recordings had to be biased to VIP neurons in more rostral parts of the IC where the commissure was clearly visible, a bias that was not present in the CRACM recordings. Also, we cannot exclude that antidromic spikes may have influenced PSP analysis. Both may account for some of the observed differences in PSP kinetics. Overall, however, these data suggest that the kinetics of Chronos-evoked PSPs were generally similar to those of electrically-evoked PSPs.

## VIP neurons in the ICc receive synaptic input from the DCN

The DCN provides one of the major sources of excitatory input to the IC (*Adams, 1979*; *Brunso-Bechtold et al., 1981*; *Oliver, 1984*; *Osen, 1972*; *Ryugo et al., 1981*). A previous study has shown that DCN afferents synapse onto glutamatergic and GABAergic neurons in the IC (*Chen et al., 2018*), but it is not known which specific classes of IC neurons receive input from the DCN. In addition, the physiology of DCN afferent synapses remains unknown. To test whether VIP neurons receive synaptic input from DCN principal neurons, we injected the right DCN with the AAV1.Syn.Chronos-GFP.WPRE.bGH viral vector and recorded from VIP neurons in the contralateral (left) ICc (*Figure 8A*, left). To confirm selective transfection of the DCN, we sliced the brainstem of every animal used and determined whether GFP expression was present and limited to the right DCN. If there was no transfection or if there was considerable expression of GFP in the auditory nerve or VCN, no recordings were performed. In most cases, GFP expression was limited to the DCN (*Figure 8A*, right) and GFP-labeled axons were present in the left ICc.

To block spontaneous IPSPs, we performed DCN CRACM experiments with GABAergic and glycinergic blockers in the bath (5 µm gabazine and 1 µM strychnine, *Figure 8B*). We found that 2–5 ms pulses of blue light elicited EPSPs in 19 of 25 neurons tested, confirming that ICc VIP neurons receive synaptic input from the DCN. Light-evoked EPSPs had moderate amplitudes (2.85 mV ± 2.98 mV) and relatively slow rise times (4.2 ms ± 1.3 ms), halfwidths (20.6 ms ± 14.4 ms) and decay time constants (22.0 ms ± 6.7 ms) (n = 6 cells, *Figure 8B*, left and 8C). Because EPSP kinetics were relatively slow, we hypothesized that DCN synapses activate NMDA receptors on VIP neurons. Interestingly, D-AP5 had no significant effect on any of the measured properties (amplitude: 2.80 mV ± 2.54 mV, p = 0.96, rise time: 4.4 ms ± 1.7 ms, p = 0.95, halfwidth: 15.4 ms ± 6.1 ms, p = 0.37, decay time constant: 29.0 ms ± 7.1 ms, p = 0.16, two tailed t-test, critical p value with Bonferroni correction for multiple comparisons = 0.0125) (*Figure 8B*, middle, and 8C). In contrast to commissural inputs where D-AP5 had a significant effect on EPSP kinetics, this suggests that DCN inputs to ICc VIP neurons do not activate NMDA receptors under resting conditions. Subsequent addition of NBQX completely abolished DCN-evoked EPSPs (*Figure 8B*, right), confirming that DCN synapses activate AMPA receptors on ICc VIP neurons. Since fusiform cells in the DCN often fire at rates exceeding 100 Hz (*Davis et al., 1996*; *Ma and Brenowitz, 2012*; *Nelken and Young, 1994*; *Spirou and Young, 1991*; *Young and Brownell, 1976*), the slow kinetics of DCN-evoked EPSPs suggests that these EPSPs undergo temporal summation in VIP neurons.

To test whether Chronos-elicited synaptic release from DCN terminals was driven by action potentials as opposed to direct activation of DCN terminals by light, we repeated the DCN CRACM experiments in the presence of 1 µM tetrodotoxin (TTX). After TTX wash-in, EPSPs elicited by blue light flashes were completely and reversibly abolished in 3 of 3 tested VIP neurons (data not shown). This suggests that synaptic release in CRACM experiments was driven by action potentials, or was at least dependent on sodium channel activation in synaptic terminals.

To determine whether PSPs evoked by DCN inputs to VIP neurons differed from those evoked in non-VIP neurons, we recorded from non-fluorescent neurons in the ICc of VIP-IRES-Cre x Ai14 mice in which the DCN was transfected with Chronos. Recordings were targeted to neurons in regions of the ICc where Chronos-GFP-labeled axons were clearly visible. We found that 8 out of 12 non-VIP neurons received input from the DCN. EPSPs evoked by DCN inputs to non-VIP neurons (n = 8) had similar amplitudes, rise time, halfwidth and decay times (amplitude: 1.7 mV ± 0.6 mV, p = 0.34; rise time: 4.7 ms ± 1.6 ms, p = 0.52; halfwidth: 16.5 ms ± 9.6 ms, p = 0.77; decay time constant 25.9 ms ± 17.8 ms, p = 0.72; two tailed t-test, critical p value with Bonferroni correction for multiple comparisons = 0.0125) compared to those recorded from VIP neurons. Although the means were not statistically different, DCN-evoked EPSPs in VIP neurons showed less variability in halfwidth and decay time than EPSPs in non-VIP neurons (*Figure 8F*). These results suggest that ICc neurons located near DCN afferents have a high probability of receiving input from the DCN, consistent with predictions made from anatomical studies (*Oliver, 1984*; *Oliver et al., 1997*).

Together, these results identify VIP neurons in the ICc as a distinct postsynaptic target of DCN afferents. Given the number of conditions that must be met for a long-range CRACM experiment to succeed, our observation that the connection probability for DCN to VIP projections was 76% (19 of 25 neurons) suggests that most VIP neurons in the ICc receive input from the DCN.

## DCN afferents drive local feedforward inhibition onto VIP neurons in the ICc

We next repeated the DCN-CRACM experiments without GABA$_A$ and glycine receptor antagonists in the bath. Under these conditions, we observed that an IPSP was elicited 2–3 ms after the onset of the light-evoked EPSP (*Figure 8D*). This IPSP could vary in strength between recorded VIP neurons. In some instances, the IPSP slightly altered the halfwidth and decay time constant of the EPSP. In other cases, the IPSP strongly limited the EPSP duration and generated a hyperpolarization after the EPSP (*Figure 8D*, left). Washing in gabazine restored the EPSP to values comparable to the EPSPs recorded in the presence of inhibitory receptor antagonists (compare *Figure 8B,C*). Washing out gabazine restored the IPSP and limited EPSP halfwidth again (*Figure 8D*, right). Across five ICc VIP neurons, the IPSP significantly shortened the halfwidth of the elicited EPSP (p = 0.048, paired t-test, *Figure 8E*). Halfwidth reductions ranged from 22% to 73%, with a median reduction of 36%. Because DCN to IC projections are glutamatergic (*Ito and Oliver, 2010*; *Oliver, 1984*), and the periolivary nuclei and nuclei of the lateral lemniscus, the sources of ascending GABAergic input to the IC (*González-Hernández et al., 1996*), were not present in the brain slice, this inhibition must be due to DCN afferents activating a local feedforward inhibitory circuit within the IC. The latency to the IPSP onset also supports the theory of a disynaptic, local inhibitory circuit, as the IPSP always succeeded the initial EPSP. Thus, these results indicate that ascending input from the DCN activates a feedforward inhibitory circuit within the IC and that this circuit regulates the duration of DCN-evoked excitation in ICc VIP neurons.

## Discussion

By combining molecular genetics with physiological and anatomical approaches, we identified VIP neurons as a novel class of glutamatergic principal neurons in the auditory midbrain. In contrast to the diverse properties present in the broader population of IC neurons, VIP neurons consistently exhibited sustained firing patterns, spiny dendrites, and stellate morphology. Surprisingly, the identification of VIP neurons revealed that a molecularly-defined class of IC neurons can broadcast the output of the IC to multiple auditory and non-auditory brain regions (*Figure 9A*). The identification of VIP neurons also enabled us to identify microcircuit motifs within the IC. In particular, we found that VIP neurons receive ascending input from the contralateral DCN and commissural input from the contralateral IC (*Figure 9B*). That VIP neurons receive input from the DCN was consistent with the distribution of VIP neurons in the ICc, which was biased toward more caudal regions where DCN afferents are prominent. Input from the DCN also drove feedforward inhibition that could sculpt the effects of excitatory inputs to VIP neurons. Thus, feedforward inhibitory circuits within the IC may regulate the temporal summation of ascending inputs. By integrating multiple sources of input and

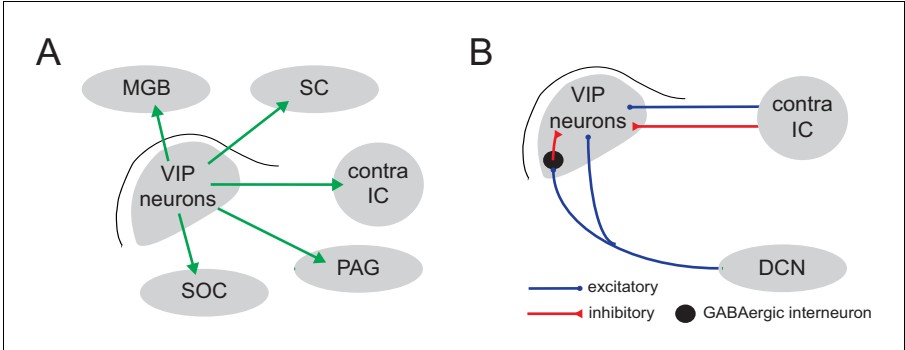

**Figure 9.** Summary of inputs and projection targets of VIP neurons in mouse IC. (**A**) Summary of the major projection targets of VIP neurons identified by axonal tract tracing: MGB (medial geniculate body), SC (superior colliculus), contralateral IC, PAG (periaqueductal gray), SOC (superior olivary complex). (**B**) Summary of the sources of input to VIP neurons identified by CRACM experiments.
DOI: https://doi.org/10.7554/eLife.43770.019

participating in most of the major projections from the IC, VIP neurons are well-positioned to broadly influence auditory computations in numerous brain regions.

## VIP neurons are a distinct class of IC neurons

It has long been argued that the classification of neurons requires a multifaceted analysis of morphological and physiological features (*Tyner, 1975*). More recent efforts have emphasized the importance of combining these features with molecular markers (*Ascoli et al., 2008*; *Tremblay et al., 2016*; *Zeng and Sanes, 2017*). This combination has proven to be particularly effective for unambiguously classifying neuron types. In the cerebral cortex, a multifaceted classification scheme including molecular markers has enabled investigations into how specific classes of interneurons shape circuit computations, sensory processing, and animal behavior (*Cichon et al., 2017*; *Kato et al., 2017*; *Kato et al., 2015*; *Kuchibhotla et al., 2017*; *Lee et al., 2013*; *Milstein et al., 2015*; *Pfeffer et al., 2013*; *Pi et al., 2013*). Similar approaches have succeeded in the amygdala, hypothalamus, basal ganglia, and other brain regions where it was previously difficult to identify neuron types (*Campbell et al., 2017*; *Capogna, 2014*; *Wallace et al., 2017*).

In the auditory midbrain, previous efforts to identify neuron classes relied on in vitro physiology, in vivo physiology, morphology, neurochemical markers, or some combination of these (*Beebe et al., 2016*; *Fujimoto et al., 2017*; *Malmierca et al., 1993*; *Oliver and Morest, 1984*; *Ono et al., 2005*; *Palmer et al., 2013*; *Peruzzi et al., 2000*; *Ramachandran et al., 1999*; *Schofield and Beebe, 2019*; *Sivaramakrishnan and Oliver, 2001*). Despite these and other attempts, the only neuron class that has been consistently identified in the IC are the large GABAergic neurons (*Beebe et al., 2016*; *Geis and Borst, 2013*; *Ito et al., 2015*; *Ito et al., 2009*; *Ito and Oliver, 2014*). However, there are currently no known molecular markers for large GABAergic neurons, making it difficult to test the role of these neurons in auditory computations (*Schofield and Beebe, 2019*).

Using a multifaceted approach, we identified VIP neurons as a novel class of IC neurons. VIP neurons share a common set of molecular, neurochemical, morphological, and physiological features. VIP neurons are labeled in the VIP-IRES-Cre mouse line and are glutamatergic. Ninety-three percent of the ICc VIP neurons in our data set had a stellate morphology and 94% of all IC VIP neurons had dendritic spines. Similarly, 90% of VIP neurons had sustained firing patterns. Although the input resistance, membrane time constant, and expression of $I_h$ varied within the population of VIP neurons, for VIP neurons in the ICc, we found that a portion of this variability reflected their location along the tonotopic axis of the ICc. Tonotopic variation in the intrinsic physiology of VIP neurons was consistent with VIP neurons having faster membrane properties and therefore better temporal coding potential in lower frequency regions of the ICc. This parallels an in vivo study that found a gradient in the temporal coding capacity of cat ICc neurons that varied from better temporal coding in low frequency regions to worse temporal coding in high frequency regions (*Rodríguez et al., 2010*).

It is important to note that it would not be possible to identify VIP neurons based on their morphology or physiology alone. VIP neurons are not the only stellate neurons in the IC, nor are they the only neurons with sustained firing patterns or dendritic spines. These results provide insight to why it has been difficult to classify neuron types in the IC. We propose that a multidimensional approach incorporating molecular markers will be essential to identifying the remaining neuron classes in the IC.

## Diverse projection patterns of VIP neurons

We found that VIP neurons project not only to MGB and contralateral IC, the most common recipients of IC projections, but also to the nucleus of the brachium of the IC, superior colliculus, periaqueductal gray, superior olivary complex, and ipsilateral IC (*Figure 6*). The number of extrinsic targets reached by VIP axons was a surprise given the relatively small population of VIP neurons. Do individual VIP neurons project to multiple targets? Previous retrograde labeling studies suggest that some patterns of collateral projection are more common than others for IC cells. IC cells that project to the contralateral thalamus appear to quite commonly have a collateral projection to the ipsilateral thalamus (*Mellott et al., 2019*). In contrast, very few IC neurons project to the thalamus and the cochlear nucleus (*Coomes and Schofield, 2004*; *Hashikawa and Kawamura, 1983*;

Okoyama et al., 2006), or to the left and right cochlear nuclei (*Schofield, 2001*). Whether IC commissural cells can have collateral projections to the thalamus has been supported (*González-Hernández et al., 1996*) or denied (*Okoyama et al., 2006*). Because retrograde tracing studies underestimate collateral projections (*Schofield et al., 2007*), such studies may have missed VIP neurons with collateral projections.

Alternatively, individual VIP neurons might project to one or a few targets. It would then be possible to subdivide VIP neurons based on their axonal projections. This would parallel the cerebral cortex, where the major classes of interneurons contain subclasses that often differ in their axonal targeting (*Tremblay et al., 2016*). If such is the case, then the unifying feature of VIP neurons might be that they perform similar computational roles within circuits, even when the circuits themselves are involved in different functions. In any event, the axonal projection patterns of individual VIP neurons, to extrinsic targets and within the IC, will be important features for further characterizing VIP subclasses and their functional roles.

## Integration of synaptic input by VIP neurons

Our results show that VIP neurons receive input from at least four sources: principal neurons in the DCN, local inhibitory neurons, and excitatory and inhibitory neurons in the contralateral IC (*Figure 8G*). In multiple instances, we observed that VIP neurons received input from the DCN and a local inhibitory neuron or a combination of excitatory and inhibitory commissural input. Given that optogenetic circuit mapping experiments underestimate connection probabilities (not all synapses are transfected by the virus), these results suggest that many individual VIP neurons integrate input from ascending, local, and commissural sources. This is consistent with previous studies showing that individual IC neurons can integrate input from numerous sources (*Ito et al., 2015*; *Ito and Oliver, 2014*).

Excitatory postsynaptic responses in IC neurons often involve activation of NMDA receptors (*Ma et al., 2002*; *Wu et al., 2004*). Under in vivo conditions, the activation of NMDA receptors can influence how IC neurons respond to tones (*Sanchez et al., 2007*). In VIP neurons, we found that excitatory commissural synapses activated AMPA and NMDA receptors, while synaptic input from DCN afferents activated only AMPA receptors. The activation of NMDA receptors occurred even though our ACSF contained 1 mM $Mg^{2+}$ and neurons were at their resting membrane potential. Interestingly, previous studies have shown that NMDA receptors in some IC neurons can be activated under similar conditions, even when AMPA are receptors blocked (*Ma et al., 2002*; *Sivaramakrishnan and Oliver, 2006*). The activation of NMDA receptors in our recordings may indicate that excitatory commissural synapses tend to be located on the distal dendrites or dendritic spines of VIP neurons, where the local membrane potential might be sufficiently depolarized by activation of AMPA receptors to remove $Mg^{2+}$ block of NMDA receptors. A distal location would be consistent with the proposed modulatory role for commissural inputs (*Orton et al., 2016*; *Orton and Rees, 2014*). Similarly, the lack of NMDA receptor activation by DCN afferents might indicate that DCN synapses are located on proximal dendrites or possibly on the soma itself, or that these synapses lack NMDA receptors. These synaptic arrangements may have important implications for auditory coding and synaptic plasticity mechanisms in VIP neurons.

In many brain regions, feedforward inhibitory circuits control the time window for temporal integration of synaptic input (*Gabernet et al., 2005*; *Pouille and Scanziani, 2001*; *Roberts et al., 2013*; *Stokes and Isaacson, 2010*). In the ICc, it was recently shown that GABAergic neurons provide local inhibitory input mainly to neurons in the same isofrequency lamina (*Sturm et al., 2014*). However, the conditions that recruit local inhibition have remained unclear. Our data provide direct evidence that activation of DCN afferents can elicit both direct excitatory input and disynaptic feedforward inhibition to VIP neurons. Feedforward inhibition can dramatically reduce EPSP halfwidth, suggesting that local feedforward inhibition regulates the temporal summation of synaptic inputs. In addition, while DCN afferents elicited modest EPSPs in VIP neurons, DCN input presumably drove spiking in the GABAergic neurons that provided feedforward inhibition. It will be important for future studies to identify this population of GABAergic neurons and the extent of their influence on auditory computations in VIP and other IC neurons.

## The distribution and sources of input to VIP neurons suggest they specialize in monaural processing

Ascending projections from most auditory brainstem nuclei restrict their synapses to subregions of the ICc, dividing the ICc into functional zones that may be specialized for processing specific classes of auditory cues (*Brunso-Bechtold et al., 1981*; *Cant, 2013*; *Cant and Benson, 2006*; *Loftus et al., 2010*; *Oliver, 2005*; *Oliver et al., 1997*). We found that VIP neurons in the ICc were distributed with a significant bias toward the caudal ICc (*Table 2*) and a tendency to be enriched in medial and dorsal regions of the ICc (*Figure 1*). Intriguingly, this distribution overlaps with the monaural domain of the ICc, an area that predominantly receives input from monaural nuclei, including the contralateral DCN and AVCN and the ipsilateral ventral nucleus of the lateral lemniscus (VNLL) (*Cant and Benson, 2006*; *Loftus et al., 2010*). In addition, a recent study in mice showed that excitatory ICc neurons that receive input from the DCN are unlikely to receive input from other ascending auditory sources but likely to receive input from the ipsilateral auditory cortex (*Chen et al., 2018*). These results lead us to hypothesize that VIP neurons are specialized for processing monaural cues, and in particular, monaural cues from the DCN. In future studies, it will be important to determine whether ascending input to VIP neurons is predominantly from the DCN, from a combination of monaural brainstem nuclei, or from both monaural and binaural sources. It will also be important to assess whether VIP neurons receive descending input from the auditory cortex and how commissural and cortical inputs shape sound processing in VIP neurons.

## VIP neurons might regulate the excitability of MGB and SOC neurons through VIP signaling

An important question is whether IC VIP neurons use VIP signaling to modulate the activity of their postsynaptic targets. In cerebral cortex, 99.1% of neurons labeled in the VIP-IRES-Cre mouse immunostained with an antibody against VIP, suggesting that Cre expression in the VIP-IRES-Cre mouse is well correlated with the production of VIP (*Prönneke et al., 2015*). There is therefore a strong likelihood that VIP neurons in the IC produce VIP. VIP acts by binding to one of three G-protein coupled receptors, VPAC1, VPAC2, and PAC1 (*Dickson and Finlayson, 2009*; *Vaudry et al., 2009*). Anatomical studies suggest that all three VIP receptors are expressed in the MGB and SOC (*Joo et al., 2004*; *Sheward et al., 1995*), both of which are postsynaptic targets of VIP neurons (*Figure 6*). Activation of VIP receptors can have significant effects on neuronal excitability. In somatosensory thalamus, activation of VIP receptors increased the activation of HCN channels, depolarizing thalamocortical neurons and inducing them to switch from burst firing to tonic firing (*Lee and Cox, 2003*; *Sun et al., 2003*). These data suggest that IC VIP neurons may potently modulate the activity of MGB neurons and other postsynaptic target neurons through VIP signaling. Thus, VIP neurons in the IC may exert an outsize influence on their long-range postsynaptic targets.

## Materials and methods

**Key resources table**

| Reagent type (species) or resource | Designation | Source or reference | Identifiers | Additional information |
|---|---|---|---|---|
| Strain, strain background (*Mus musculus*) | C57BL/6J | The Jackson Laboratory | JAX:000664 | |
| Genetic reagent (*Mus musculus*) | VIP-IRES-Cre | The Jackson Laboratory | JAX:010908 | |
| Genetic reagent (*Mus musculus*) | Ai14 | The Jackson Laboratory | JAX:007914 | |
| Antibody | anti-GAD67 (mouse monoclonal) | Millipore | RRID:AB_2278725 Cat#:MAB5406 | IHC (1:1000) |
| Antibody | anti-NeuN (rabbit polyclonal) | Millipore | RRID:AB_10807945 Cat#:ABN78 | IHC (1:500) |

*Continued on next page*

*Continued*

| Reagent type (species) or resource | Designation | Source or reference | Identifiers | Additional information |
|---|---|---|---|---|
| Antibody | anti-bNOS (mouse monoclonal) | Sigma-Aldrich | RRID:AB_260754 Cat#:N2280 | IHC (1:1000) |
| Antibody | anti-mouse IgG Alexa Fluor 488 (donkey polyclonal) | ThermoFisher | RRID:AB_141607 Cat#:A-21202 | IHC (1:500) |
| Antibody | anti-rabbit IgG Alexa Fluor 488 (donkey polyclonal) | ThermoFisher | RRID:AB_2535792 Cat#:A-21206 | IHC (1:500) |
| Recombinant DNA reagent | AAV1.Syn.Chronos-GFP.WPRE.bGH | University of Pennsylvania Vector Core/Addgene | Addgene:59170-AAV1 RRID:Addgene_59170 | http://n2t.net/addgene:59170 |
| Recombinant DNA reagent | AAV1.CAG.FLEX. eGFP.WPRE.bGH | University of Pennsylvania Vector Core/Addgene | Addgene:51502-AAV1 RRID:Addgene_51502 | http://n2t.net/addgene:51502 |
| Chemical compound, drug | gabazine | Hello Bio | Cat#:HB0901 | also called SR95531 hydrobromide |
| Chemical compound, drug | strychnine hydrochloride | Sigma-Aldrich | Cat#:S8753 | |
| Chemical compound, drug | D-AP5 | Hello Bio | Cat#:HB0225 | |
| Chemical compound, drug | NBQX disodium salt | Hello Bio | Cat#:HB0443 | |
| Software, algorithm | Igor Pro 7 and 8 | Wavemetrics | RRID:SCR_000325 | |
| Software, algorithm | MATLAB R2018a and R2018b | Mathworks | RRID:SCR_001622 | |
| Software, algorithm | Neurolucida | MBF Bioscience | RRID:SCR_001775 | |
| Software, algorithm | Neurolucida 360 | MBF Bioscience | RRID:SCR_016788 | |

## Animals

All experiments were approved by the University of Michigan Institutional Animal Care and Use Committee and were in accordance with NIH guidelines for the care and use of laboratory animals. Animals were kept on a 12 hour day/night cycle with ad libitum access to food and water. VIP-IRES-Cre mice (*Vip^tm1(cre)Zjh*/J, Jackson Laboratory, stock # 010908) (*Taniguchi et al., 2011*) were crossed with Ai14 reporter mice (B6.Cg-*Gt(ROSA)26Sor^tm14(CAG-tdTomato)Hze*/J, Jackson Laboratory, stock # 007914) (*Madisen et al., 2010*) to yield F1 offspring that expressed the fluorescent protein tdTomato in VIP neurons. For control experiments, C57BL/6J mice (Jackson Laboratory, stock # 000664) were used. Because mice on the C57BL/6 background undergo age-related hearing loss, experiments were restricted to an age range where hearing loss should be minimal or not present (*Zheng et al., 1999*). This age range was P70 or less for all mice except for three C57BL/6J mice used for electrophysiology experiments that were aged P89, P90, and P113.

## Immunohistochemistry

Mice were deeply anesthetized and perfused transcardially with 0.1 M phosphate-buffered saline (PBS), pH 7.4, for 1 min and then with a 10% buffered formalin solution (Millipore Sigma, cat# HT501128) for 10 min. Brains were collected and post-fixed in the same fixative for 2 hr and cryoprotected overnight at 4°C in 0.1 M PBS containing 20% sucrose. Brains were cut into 40 µm sections on a vibratome or freezing microtome. Sections were rinsed in 0.1 M PBS, and then treated with 10% normal donkey serum (Jackson ImmunoResearch Laboratories, West Grove, PA) and 0.3% Triton X-100 for 2 hr. Slides were incubated overnight at 4 °C in mouse anti-GAD67 (1:1000; Millipore Sigma, cat# MAB5406), rabbit anti-NeuN (1:500; Millipore Sigma, cat# ABN78), or mouse anti-bNOS

(1:1000; Millipore Sigma, cat# N2280). The next day, sections were rinsed in 0.1 M PBS and incubated in Alexa Fluor 488-tagged donkey anti-mouse IgG or donkey anti-rabbit IgG (1:500, Thermo Fisher, cat# A-21202 and A-21206) for 1 hr at room temperature. Sections were then mounted on gelatin-subbed slides (SouthernBiotech, cat# SLD01-BX) and coverslipped using Fluoromount-G (SouthernBiotech, cat# 0100–01). Images were collected using a 1.30 NA 40x oil-immersion objective or a 1.40 NA 63x oil-immersion objective on a Leica TCS SP8 laser scanning confocal microscope.

## Antibody validation

The mouse monoclonal anti-GAD67 antibody (Millipore Sigma, cat# MAB5406) was raised against the 67 kDA isoform of glutamic acid decarboxylase (GAD). The manufacturer reports that Western blot analysis shows no cross-reactivity with the 65 kDa isoform of GAD (GAD65). This antibody has been previously used to identify GABAergic cells in the IC (*Beebe et al., 2016*; *Ito et al., 2009*; *Mellott et al., 2014*). The mouse monoclonal anti-nitric oxide synthase-brain (bNOS) (Sigma, cat# N2280) was raised against the IgG1 isotype from the NOS-B1 hybridoma. The manufacturer reports that anti-bNOS reacts specifically with nitric oxide synthase (NOS), derived from brain (bNOS, 150 – 160 kDa). This antibody has been previously used, in guinea pig and mouse, to delineate the borders of the IC (*Coote and Rees, 2008*; *Keesom et al., 2018*). To perform NeuN staining, we used a rabbit polyclonal antibody (Millipore Sigma, cat# ABN78). The manufacturer reports that anti-NeuN specifically recognizes the DNA-binding, neuron-specific protein NeuN, which is present in most central and peripheral neuronal cell types of all vertebrates tested. Previous studies reported the use of this antibody to label neurons in the IC (*Beebe et al., 2016*; *Foster et al., 2014*; *Mellott et al., 2014*).

## Analysis of GAD67 staining

Images from representative sections of the IC (n = 3 animals, two sections per animal, one caudal and one middle) were collected at 2 µm depth intervals with a 1.30 NA 40x oil-immersion objective on a Leica TCS SP8 laser scanning confocal microscope. Images were analyzed using Fiji software (*Rueden et al., 2017*; *Schindelin et al., 2012*). Consistent with previous studies, we found that the anti-GAD67 antibody did not penetrate the entire depth of the tissue sections (*Beebe et al., 2016*; *Mellott et al., 2014*). We therefore restricted our analysis to the top 10 – 12 µm of each section, where the antibody was fully penetrant. Within this region, we manually marked every GAD67$^+$ cell body and every tdTomato$^+$ cell body in the left IC. The green (GAD67) and red (tdTomato) color channels were analyzed separately, so that labeling in one channel did not influence analysis of the other channel. After cells were marked, the GAD67 and tdTomato color channels were merged, and every instance where a cell body contained markers for both GAD67 and tdTomato was counted. The number of double-labeled cells was compared to the total number of tdTomato$^+$ neurons to determine the percentage of tdTomato$^+$ neurons that were GAD67$^+$.

## Analysis of NeuN staining with design-based stereology

A design-based stereology approach was used to estimate the numbers of NeuN$^+$ and tdTomato$^+$ neurons in anti-NeuN stained sections (*Schmitz and Hof, 2005*). To collect systematic random samples, a virtual 370 µm x 370 µm grid was overlaid on the IC section. The starting coordinates for the grid were set using the Mersenne Twister random number generator in Igor Pro 7 or 8 (WaveMetrics Inc). Images were then collected at coordinates determined by the upper-left intersection of each grid-square that fell over the left IC. Each image consisted of a 184 µm x 184 µm Z-stack collected at 1 µm depth intervals with a 1.40 NA 63x oil immersion objective on a Leica TCS SP8 confocal microscope. Six to sixteen images were collected per slice. Three slices were analyzed per mouse, with slices from each mouse evenly distributed along the rostral-caudal axis of the IC. Images were imported to Neurolucida 360 (MBF Bioscience), where neurons were counted using the optical fractionator approach (*West et al., 1991*). In this approach, we determined the image planes corresponding to the top, center, and bottom of the slice in each image stack. Top and bottom regions of each slice (≥2 µm thick) were treated as guard zones and discarded from subsequent analysis. Removal of guard zones left a 15 µm-thick region at the center of the slice for subsequent analysis. Neurons within this region were counted by making a single mark at the top of each cell. Cells crossing the right and top borders of the image stack were counted, whereas those crossing the left and

bottom borders were not. The green (NeuN) and red (tdTomato) color channels were analyzed separately, so that labeling in one channel did not affect analysis of the other. Next, the color channels were merged and cells with both NeuN and tdTomato markers were counted. In every instance, tdTomato$^+$ cells were also NeuN$^+$ (208/208 cells). The total number of double-labeled (tdTomato$^+$/NeuN$^+$) cells was then compared to the total number of NeuN$^+$ cells.

## Analysis of the distribution of VIP neurons

Following transcardial perfusion as described previously, brains from three VIP-IRES-Cre x Ai14 mice were frozen and sectioned on a sliding microtome. Brains were cut into 40 μm sections, one each in the transverse, sagittal, and horizontal planes, and sections were collected in three series. The distribution of tdTomato-expressing (VIP$^+$) cells in one series from each case was analyzed using a Neurolucida system (MBF Bioscience, Williston, VT) attached to a Zeiss Axioimager.Z1 fluorescence microscope. Major IC subdivisions, including the central nucleus (ICc), dorsal cortex, (ICd) and lateral cortex (IClc), were identified by comparing bNOS and GAD67 immunostains with previous studies of mouse IC (*Dillingham et al., 2017*; *Meininger et al., 1986*; *Ono et al., 2016*; *Willard and Ryugo, 1983*). Neurolucida Explorer was used to export drawings to Adobe Illustrator for figure preparation.

## Electrophysiology

Mice of either sex were used, aged postnatal day (P) 21 to P70 for VIP-IRES-Cre x Ai14 crosses and P21 to P113 for C57BL/6J animals. Mice were deeply anesthetized with isoflurane, decapitated, and the brain was dissected quickly in ~34 °C artificial cerebrospinal fluid (ACSF) containing (in mM): 125 NaCl, 12.5 Glucose, 25 NaHCO$_3$, 3 KCl, 1.25 NaH$_2$PO$_4$, 1.5 CaCl$_2$, 1 MgSO$_4$, bubbled to a pH of 7.4 with 5% CO$_2$ in 95% O$_2$. Chemicals were obtained from Fisher Scientific or Millipore Sigma unless stated otherwise. Coronal or parasagittal slices (200 – 250 μm) containing the IC were cut in ~34 °C ACSF with a vibrating microtome (VT1200S, Leica Biosystems) and incubated at 34 °C for 30 min in a holding chamber filled with ACSF and bubbled with 5% CO$_2$ in 95% O$_2$. After incubation, slices were stored at room temperature until used for recordings.

To make recordings, slices were placed in a recording chamber under a fixed stage upright microscope (BX51WI, Olympus Life Sciences) and were constantly perfused with 34 °C ACSF at ~2 ml/min. All recordings were conducted near physiological temperature (34 °C). IC neurons were patched under visual control using infrared Dodt gradient contrast and epifluorescence imaging. Recordings were performed with a BVC-700A patch clamp amplifier (Dagan Corporation). Data were low pass filtered at 10 kHz, sampled at 50 kHz with a National Instruments PCIe-6343 data acquisition board, and acquired using custom written algorithms in Igor Pro. For every recording, series resistance and pipette capacitance were corrected using the bridge balance circuitry of the BVC-700A. Recordings with a series resistance above 25 MΩ were discarded. All membrane potentials have been corrected for a liquid junction potential of 11 mV.

Electrodes were pulled from borosilicate glass (outer diameter 1.5 mm, inner diameter 0.86 mm, Sutter Instrument) to a resistance of 3.5 – 4.5 MΩ using a P-1000 microelectrode puller (Sutter Instrument). The electrode internal solution contained (in mM): 115 K-gluconate, 7.73 KCl, 0.5 EGTA, 10 HEPES, 10 Na$_2$ phosphocreatine, 4 MgATP, 0.3 NaGTP, supplemented with 0.1% biocytin (w/v), pH adjusted to 7.3 with KOH and osmolality to 290 mmol/kg with sucrose.

Input resistance was determined by delivering a series of 100 ms hyperpolarizing current steps incremented to elicit hyperpolarization ranging from just below the resting membrane potential to < −110 mV. For each response, the amplitudes of the peak (most negative value) and steady-state (average of last 10 ms of response) hyperpolarization were measured relative to the resting potential. Voltage versus current plots were prepared, and input resistance was determined from the slopes of lines fit to the peak ($R_{pk}$) and steady-state ($R_{ss}$) data for current steps that achieved a peak hyperpolarization between 0 and −15 mV relative to rest. Membrane time constant was determined by delivering 50 current steps at an amplitude that hyperpolarized the membrane potential by 1–3 mV. Current step duration was set to ensure that the membrane potential achieved a steady-state value before the end of the current step. An exponential function was then fit to onset of each response and the median time constant determined.

For electrical stimulation of commissural inputs, a glass electrode filled with ACSF was placed into the commissure near the midline, connected to a DS3 constant current, isolated stimulator (Digitimer Ltd.). PSPs were elicited with 100 µs electric shocks, ranging from 45 µA to 320 µA.

To isolate or manipulate synaptic events, the following pharmacological agents were used, all diluted in standard ACSF: 5 µM SR95531 (gabazine, $GABA_A$ receptor antagonist, Hello Bio), 1 µM strychnine hydrochloride (glycine receptor antagonist, Millipore Sigma), 50 µM D-AP5 (NMDA receptor antagonist, Hello Bio), 10 µM NBQX disodium salt (AMPA receptor antagonist, Hello Bio), 1 µM TTX (voltage dependent sodium channel blocker, Hello Bio).

Data analysis was performed using custom written algorithms in Igor Pro or MATLAB (Mathworks). Statistical tests were performed in R Studio (R Studio, Boston) for R 3.5.1 (The R Project for Statistical Computing, The R Foundation).

## Post hoc reconstructions of morphology and morphology analysis

After recordings, the electrode was removed slowly to allow the cell membrane to reseal, and the slice was fixed in 4% paraformaldehyde (PFA) in 0.1 M phosphate buffer (PB, pH 7.4) for 12 – 24 hr. Slices were then washed in 0.1 M PB and stored in 0.1 M PB for up to three weeks. Recorded neurons were stained using fluorescent biocytin-streptavidin histochemistry. In brief, slices were washed in 0.1 M PB three times for 10 min (3 × 10 min in PB), permeabilized in 0.2% Triton X-100 in 0.1 M PB for 2 hr, washed 3 × 10 min in PB, and stained at 4 ˚C for 48 hr with streptavidin-Alexa Fluor 488 or 647, diluted 1:1000 in 0.1 M PB. Slices were then washed 3 × 10 min in PB and mounted on Superfrost Plus microscope slides in anti-fade media (Fluoromount-G). Z-stack images of streptavidin-Alexa Fluor labeled cells were obtained with a Leica TCS SP8 laser scanning confocal microscope using a 1.40 NA 63x oil-immersion objective. Three-dimensional reconstructions of neuronal morphology and quantitative analyses of soma and dendrite shape were performed on image stacks imported into Neurolucida 360 (MBF Bioscience). To facilitate comparisons of neuronal morphology, all reconstructed neurons are displayed as if they were in the left IC as viewed from a caudal perspective. Reconstructions of neurons that were located in the right IC were flipped along the dorsal-ventral axis so that they appear as if they were in the left IC.

## Correlation of neuron location and intrinsic physiology

Following biocytin-streptavidin histochemistry, tile scan images of the entire IC were collected using a 20x objective on a Leica TCS SP8 confocal microscope. These images were then used to determine medial-lateral and dorsal-ventral coordinates of recorded neurons. The medial-lateral coordinate was measured as the distance of the soma from the medial axis (midline) of the IC slice (x axis). The dorsal-ventral coordinate was measured as the distance of the soma from the dorsal-most edge of the IC slice (y axis). Neurons in the right IC were combined with those from the left IC by multiplying the medial-lateral coordinate of neurons from the right IC by −1. Neurons were assigned to IC subdivisions, including the central nucleus (ICc), dorsal cortex, (ICd) and lateral cortex (IClc), by comparing neuron location in 20x tile scans to IC subdivision borders determined from standard series of IC sections immunostained for bNOS or GAD67 (*Choy Buentello et al., 2015*; *Coote and Rees, 2008*). Neuron coordinates were then compared to physiological parameters obtained during whole cell recordings. To test for correlations, data were fit with a plane using the Levenberg-Marquardt least squares method in Igor Pro. Fit quality was assessed with Pearson's correlation coefficient and the adjusted $R^2$. Fit significance (p value) was calculated based on the chi-squared statistic from the fit and the chi-squared cumulative distribution function.

## Determination of major axes of neuron morphology

Neuron morphology was reconstructed as described above. To determine the 'length' and 'width' axes of the dendritic arbors, the set of coordinates describing the morphology of the dendritic arbor of each neuron was exported from Neurolucida 360 (MBF Bioscience). Coordinates were imported to Igor Pro, where principal components analysis (PCA) was performed on either the x and y coordinates (2D PCA) or the x, y, and z coordinates (3D PCA). The orientation of the length and width axes was then derived from the first and second principal directions of the resulting eigenvector matrices. 2D PCA was used to determine the orientation of neurons within the coronal plane. 3D PCA was used to determine the axes to use for measuring dendritic arbor length/width ratios. For this, the

spread of the dendritic arbor along the first and second principal directions was determined by rotating each morphology coordinate set according to its eigenvector matrix, then calculating the range from the minimum to maximum coordinates along the x (length, first principal direction) and y (width, second principal direction) axes.

### Intracranial virus injections

Intracranial virus injections were performed on mice age P21 – P35 using standard aseptic techniques. Throughout the procedure, mice were anesthetized with isoflurane and their body temperature maintained with a homeothermic heating pad. An injection of the analgesic carprofen (5 mg/kg, CarproJect, Henry Schein Animal Health) was delivered subcutaneously. The scalp was shaved and a rostro-caudal incision was made along the midline to expose the skull. Injection sites were mapped using stereotaxic coordinates relative to the lambda suture. A single craniotomy was performed using a micromotor drill (K.1050, Foredom Electric Co.) with a 0.5 mm burr (Fine Science Tools).

Viral constructs were injected with a NanoJect III nanoliter injector (Drummond Scientific Company) connected to a MP-285 micromanipulator (Sutter Instruments). Glass injection pipettes were prepared by pulling capillary glass (Drummond Scientific Company) with a P-1000 microelectrode puller (Sutter Instrument). The injector tip was cut to an opening of ~20 µm and beveled at 30° with a BV-10 pipette beveller (Sutter Instrument). Injectors were back-filled with mineral oil and then front-filled with virus. AAV1.Syn.Chronos-GFP.WPRE.bGH (University of Pennsylvania Vector Core, Lot# CS1027L, 2.986e13 genome copies (GC)/ml) was used for CRACM experiments. AAV1.CAG. FLEX.eGFP.WPRE.bGH (Allen Institute 854, University of Pennsylvania Vector Core, Lot# CS0922, 4.65e13 GC/ml) was used for axonal tract tracing. For CRACM experiments, the IC was injected via two penetrations. Virus deposits were made at 250 µm intervals along the dorsal-ventral axis, resulting in four deposits in penetration one and three deposits in penetration 2. At each depth, 20 nl of virus was deposited, resulting in seven virus deposits and a total load of 140 nl virus per injected IC. DCN injections were limited to two deposits of 20 nl virus. For axonal tracing studies, viral load was reduced to 40 nl total (20 nl per site) to achieve sparser labeling of neurons. Injections were made at the coordinates shown in *Table 3*.

After injections were completed, the scalp was sutured with Ethilon 6–0 (0.7 metric) nylon sutures (Ethicon USA LLC), and the wound was treated with 0.5 – 1 ml 2% Lidocaine hydrochloride jelly (Akorn Inc). Once mice were ambulatory, they were returned to the vivarium where they were monitored daily until sutures fell out and the wound was completely healed.

### Channelrhodopsin-assisted circuit mapping

After allowing 3 – 4 weeks for Chronos expression, animals were used for in vitro slice electrophysiology experiments as described above, with the exception that after decapitation all steps were performed in red light and recordings were conducted in darkness or red light to limit Chronos activation. For standard CRACM experiments, recordings were targeted to VIP neurons. In additional control experiments, recordings were targeted to non-fluorescent IC neurons in VIP-IRES-Cre x Ai14 crosses. During whole cell recordings, Chronos was activated by brief pulses of 470 nm light emitted by a blue LED coupled to the epi-fluorescence port of the microscope and delivered to the brain slice through a 0.80 NA 40x water immersion objective with a field number of 26.5 mm. Accordingly, the blue light spot had an area of 0.345 mm$^2$. In all CRACM experiments, the soma of the recorded neuron was present in the field of view. Blue light flashes were 2 to 5 ms long,

**Table 3.**
Stereotaxic coordinates for virus injections. All coordinates are relative to the lambda suture.

| Location | X coordinate (caudal) | Y coordinate (lateral) | Z coordinates (depth) |
| --- | --- | --- | --- |
| Right IC penetration 1 (CRACM) | -900 µm | 1000 µm | 2250 - 1500 µm, 250 µm interval |
| Right IC penetration 2 (CRACM) | -900 µm | 1250 µm | 2250 - 1750 µm, 250 µm interval |
| Right IC penetration 1 (axonal tracing) | -900 µm | 1000 µm | 1850 µm, 2000 µm |
| Right DCN | -1325 µm | 2150 µm | 4750 µm, 4550 µm |

DOI: https://doi.org/10.7554/eLife.43770.020

illuminated the entire field of a 0.80 NA 40x objective, and yielded optical power densities that ranged from 6 to 48 mW/mm$^2$. Optical power was set using a minimal stimulation protocol. In general, the shortest stimulus duration that elicited a PSP was chosen, combined with 120% of the optical power that was determined as the threshold to elicit PSPs. Yet, when using a 5 ms pulse with maximum optical power, EPSP kinetics did not change significantly (8 of 8 recorded neurons, data not shown) compared to the minimal stimulation paradigm. This suggests that light pulses elicited action potentials with consistent durations, regardless of optical power. Recording sweeps with light flashes were repeated 20 to 50 times in 0.5 – 1 s intervals to obtain average PSP values. During experiments to investigate receptor contribution to PSPs, drugs were washed in for at least 10 min before recording under the new condition. For each receptor antagonist, 7 – 8 washout experiments were conducted. In each case, drug effects reversed after washout (data not shown).

## Axonal projections

The right IC of VIP-IRES-Cre x Ai14 mice was injected with AAV1.CAG.FLEX.eGFP.WPRE.bGH and transcardially perfused 3 – 4 weeks later, as described above. Brains were frozen and sectioned at 40 µm on a sliding microtome. For some brains, sections were collected serially, and for others, sections were collected in three series. The brains were examined for eGFP-labeled axons and boutons, which were interpreted as VIP$^+$ projections originating in the IC. Some sections were counterstained with a fluorescent Nissl stain (Neurotrace 640/660, ThermoFisher, cat# N21483). Injection sites comprised a collection of eGFP-labeled cell bodies. Cases were included for analysis only if the eGFP labeled cell bodies were restricted to the IC. Images were collected on a Zeiss AxioImager.Z2 microscope. High magnification images were collected as z-stacks using a 1.40 NA 63X oil-immersion objective and structured illumination (Zeiss Apotome 2) for optical sectioning at 0.5 µm intervals. Images shown are maximum projections of collected stacks. Adobe Photoshop was used to colorize images, to globally adjust levels, and to add scale bars.

## Acknowledgements

We thank Gabriel Corfas and Carol F Elias for helpful discussions and advice. This work was supported by a Deutsche Forschungsgemeinschaft Research Fellowship (GO 3060/1–1, project number 401540516, to DG), an Emerging Research Grant from the Hearing Health Foundation (MTR), and National Institutes of Health Grants R56 DC016880 (MTR) and R01 DC004391 (BRS).

## Additional information

### Funding

| Funder | Grant reference number | Author |
|---|---|---|
| Deutsche Forschungsgemeinschaft | 401540516 | David Goyer |
| Hearing Health Foundation | Emerging Research Grant | Michael Thomas Roberts |
| National Institutes of Health | DC016880 | Michael Thomas Roberts |
| National Institutes of Health | DC004391 | Brett R Schofield |

The funders had no role in study design, data collection and interpretation, or the decision to submit the work for publication.

### Author contributions

David Goyer, Brett R Schofield, Michael T Roberts, Conceptualization, Formal analysis, Funding acquisition, Investigation, Methodology, Writing—original draft, Writing—review and editing; Marina A Silveira, Conceptualization, Formal analysis, Investigation, Methodology, Writing—original draft, Writing—review and editing; Alexander P George, Formal analysis, Investigation, Methodology, Writing—review and editing; Nichole L Beebe, Formal analysis, Investigation, Methodology, Writing—original draft, Writing—review and editing; Ryan M Edelbrock, Peter T Malinski, Formal analysis, Investigation, Writing—review and editing

## Author ORCIDs

David Goyer [iD] http://orcid.org/0000-0001-7547-8285
Brett R Schofield [iD] https://orcid.org/0000-0002-0875-7759
Michael T Roberts [iD] http://orcid.org/0000-0003-2835-8752

## Ethics

Animal experimentation: All experiments were approved by the University of Michigan Institutional Animal Care and Use Committee (protocol # PRO00006642 and PRO00008773) and were in accordance with NIH guidelines for the care and use of laboratory animals.

## Decision letter and Author response

Decision letter https://doi.org/10.7554/eLife.43770.023
Author response https://doi.org/10.7554/eLife.43770.024

## Additional files

### Supplementary files

• Transparent reporting form
DOI: https://doi.org/10.7554/eLife.43770.021

### Data availability

All data generated or analysed during this study are included in the manuscript and supporting files. Source data files have been provided for Figures 3, 4, 5, 7, and 8 and Table 2.

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
