## [Decision Letter]

Thank you for submitting your article "A novel class of inferior colliculus principal neurons labeled in vasoactive intestinal peptide-cre mice" for consideration by *eLife*. Your article has been reviewed by three peer reviewers, and the evaluation has been overseen by Andrew King as the Senior Editor and Reviewing Editor. The following individuals involved in review of your submission have agreed to reveal their identity: Charles Lee (Reviewer #1); Douglas Oliver (Reviewer #2); J Gerard G Borst (Reviewer #3).

The reviewers have discussed the reviews with one another and the Reviewing Editor has drafted this decision to help you prepare a revised submission.

Summary:

Despite the importance of the inferior colliculus (IC) as a structure through which all forms of auditory information pass, previous methods have proven inadequate for identifying functionally distinct neuronal types and their circuits in this midbrain structure. In this study, the authors take advantage of molecular markers in mice to identify vasoactive intestinal peptide (VIP) neurons in the central nucleus of the IC and utilize an impressive number of methods to characterize their morphology, intrinsic membrane properties, and circuitry. They show convincingly that the VIP neurons are spiny stellate glutamatergic neurons and, as a class, project to many targets of the IC. Furthermore, these neurons show sustained firing and receive mixed inputs from the contralateral IC and a direct excitatory projection from the dorsal cochlear nucleus (DCN), followed by feedforward inhibitory inputs. Although the function of the VIP neurons is unclear from the data presented, this study provides an important contribution to our understanding of the neural subtypes in the inferior colliculus, with the VIP-cre line to be a highly useful tool to dissect auditory processing within this complex midbrain structure.

Essential revisions:

1) The paper references the recent Chen et al., 2018, manuscript. Does the current study reveal any connectional differences with the broader population of glutamatergic neurons, particularly across subdivisions of the inferior colliculus? For the CRACM experiments, the reviewers were concerned that no recordings are presented from non-VIP labeled neurons. A comparison of VIP vs non-VIP neurons, as was performed for the intrinsic physiology, would help strengthen the argument for differences between them. If this is because of technical difficulties associated with combining these recordings with optogenetic stimulation, you should test the commissural IC connections of VIP versus non-VIP neurons using conventional electrical stimulation and an in vitro commissural slice preparation (e.g. Lee et al., 2015, Hear Res).

2) The localization of the VIP neurons within the IC is interesting but raises some issues with the analysis. The images in Figure 1 show that the concentration of VIP neurons in the dorsomedial and caudal central nucleus is much higher than in other regions of IC. This suggests a distribution of VIP cells that is perpendicular to the tonotopic map, with a single isofrequency lamina comprising one region where VIP neurons are present and another where VIP neurons are sparse. A number of studies in different species have suggested similar differences in innervation from different regions of the brainstem. For example, when the distribution of inputs from dorsal cochlear nucleus (DCN) and the lateral superior olive (LSO) are compared in the cat, the DCN axons extend the full length of the lamina, including the dorsomedial part of the central nucleus, while the LSO inputs are restricted to the ventrolateral part of the lamina. Because of the uneven distribution of the VIP neurons, pooling all parts of the IC in the density measurements to determine the percentage of VIP that are glutamatergic and the percentage of IC neurons that are VIP positive is not a good approach (subsection “VIP neurons are glutamatergic and represent 1.8% of IC neurons”). The distribution of the VIP cells across the IC should be mapped and reported, and examined in relation to the location of specific IC inputs. This should help to strengthen conclusions about the possible role of the VIP neurons in auditory processing. This is also a potential problem with the percentages of neurons with different intrinsic properties (subsection “VIP neurons exhibit sustained firing patterns and their intrinsic physiology varies along the tonotopic axis of the ICc”).

3) Both the contralateral DCN and the contralateral IC are major sources of input to gerbil IC neurons (Nordeen et al., 1983; Moore et al., 1998), so it is perhaps not so surprising that this is true for mouse VIP neurons as well. Sustained firing is the most common firing pattern. The projections of the VIP neurons are investigated and documented in insufficient detail to tell whether these are special compared to other IC neurons. Overall, it seems that VIP neurons are an uncommon neuron type (<2%) with rather common properties, and, based on the present results, it is therefore not straightforward to infer their unique contribution to auditory processing by the IC. This needs to be addressed, including through the approaches suggested above.

4) The authors make the advantages of the optogenetic approach in Figure 7 over simple electrical stimulation insufficiently clear. As there is no selectivity in the labelling and little or no selectivity in the activation, what is the added value for activating the contralateral projections? The optogenetic approach also has disadvantages, e.g. how meaningful is an EPSP rise time of 4.2 ms for a 2-5 ms pulse of light? It seems that a minimal stimulation protocol was used (subsection “Channelrhodopsin-assisted circuit mapping”), but without more details on the criteria to set duration and intensity, it is hard to assess whether the PSP kinetics are meaningful. As a minimum, more details are required to assure us that release was driven by action potentials (or if this is not possible do the TTX control).

---

## [Author Response]

Essential revisions:1) The paper references the recent Chen et al., 2018, manuscript. Does the current study reveal any connectional differences with the broader population of glutamatergic neurons, particularly across subdivisions of the inferior colliculus?

We found that, as a population, VIP neurons in the central nucleus (ICc) receive excitatory synaptic input from the contralateral DCN and both excitatory and inhibitory input from the contralateral IC. Chen et al. reported little or no correlation among the glutamatergic ICc neurons receiving input from the contralateral DCN and those receiving input from the contralateral IC (Pearson’s r ≈ 0 [green color], their Figure 6B). Our data suggest that VIP neurons in the ICc are highly likely to receive input from the contralateral DCN (76% – 19 of 25 tested neurons), and moderately likely to receive input from the contralateral IC (50% of tested neurons received inhibitory input from the contralateral IC; 41% of tested neurons received excitatory input from the contralateral IC). We therefore do not think our results argue strongly for or against differences relative to the Chen et al. study. In addition, we feel that an examination of the sources of synaptic input to VIP neurons in ICd and IClc are beyond the scope of the present study but are important issues we will pursue in future work.

In the context of Essential Revision 2, we now address the Chen et al. results in more detail in the Discussion, subsection “The distribution and sources of input to VIP neurons suggest they specialize in monaural processing”. The Chen et al. study reported that excitatory ICc neurons receiving DCN input clustered away from other excitatory neurons and were unlikely to receive input from other ascending auditory sources (their Figure 6B; correlations near or less than zero). These results suggest that the main source of ascending auditory input to VIP neurons may be from the DCN.

For the CRACM experiments, the reviewers were concerned that no recordings are presented from non-VIP labeled neurons. A comparison of VIP vs non-VIP neurons, as was performed for the intrinsic physiology, would help strengthen the argument for differences between them. If this is because of technical difficulties associated with combining these recordings with optogenetic stimulation, you should test the commissural IC connections of VIP versus non-VIP neurons using conventional electrical stimulation and an in vitro commissural slice preparation (e.g. Lee et al., 2015, Hear Res).

We conducted a new CRACM experiment to test the physiology of synaptic input from the contralateral DCN to non-VIP neurons in the ICc (i.e. neurons that do not express tdTomato in VIP-IRES-Cre x Ai14 mice). Results of this experiment are shown in Figure 8F and detailed in the Results section, paragraph five of subsection “VIP neurons in the ICc receive synaptic input from the DCN”. In recordings from 12 non-VIP neurons, 8 received synaptic input from the DCN. In these neurons, the amplitude and kinetics of light-evoked EPSPs did not significantly differ from EPSPs recorded from VIP neurons. However, the halfwidths and decay time constants of EPSPs recorded from non-VIP neurons were more variable, suggesting that properties of DCN input to VIP neurons tend to be more homogenous, as might be expected for synaptic input to a defined population of neurons. Interestingly, the amplitude of DCN-evoked EPSPs was remarkably similar among VIP and non-VIP neurons (with the exception of one outlier in the VIP group). This might suggest that DCN synapses exert similar influence across their postsynaptic targets in the ICc.

A number of previous studies have examined the synaptic physiology of commissural inputs to unidentified neuron types in the IC. For example:

Reetz G, Ehret G. 1999. Inputs from three brainstem sources to identified neurons of the mouse inferior colliculus slice. Brain Res 816:527–543.

Moore DR, Kotak VC, Sanes DH. 1998. Commissural and lemniscal synaptic input to the gerbil inferior colliculus. J Neurophysiol 80:2229–2236.

Vale C, Sanes DH. 2002. The effect of bilateral deafness on excitatory and inhibitory synaptic strength in the inferior colliculus. Eur J Neurosci 16:2394–2404.

Vale C, Juíz JM, Moore DR, Sanes DH. 2004. Unilateral cochlear ablation produces greater loss of inhibition in the contralateral inferior colliculus. Eur J Neurosci 20:2133–2140. doi:10.1111/j.1460-9568.2004.03679.x

In particular, Moore et al. found that most IC neurons receive synaptic input from the contralateral IC, and they showed that these commissural inputs varied in their synaptic physiology. They further concluded that they were “unable to detect a relationship between dendritic architecture and synaptic input.” Because of these studies, we do not think it would be particularly informative to use CRACM to test the properties of commissural EPSPs and IPSPs onto non-VIP neurons. In general, we would expect to find results similar to those observed for the DCN CRACM experiment; namely that PSPs recorded from non-VIP neurons would have overlapping but more diverse kinetics compared to those recorded from VIP neurons. Overall, based on our results and previous studies from the IC and other brain regions, we think it unlikely that the properties of synaptic input to a specific neuron class will be a defining feature of that neuron class.

2) The localization of the VIP neurons within the IC is interesting but raises some issues with the analysis. The images in Figure 1 show that the concentration of VIP neurons in the dorsomedial and caudal central nucleus is much higher than in other regions of IC. This suggests a distribution of VIP cells that is perpendicular to the tonotopic map, with a single isofrequency lamina comprising one region where VIP neurons are present and another where VIP neurons are sparse. A number of studies in different species have suggested similar differences in innervation from different regions of the brainstem. For example, when the distribution of inputs from dorsal cochlear nucleus (DCN) and the lateral superior olive (LSO) are compared in the cat, the DCN axons extend the full length of the lamina, including the dorsomedial part of the central nucleus, while the LSO inputs are restricted to the ventrolateral part of the lamina. Because of the uneven distribution of the VIP neurons, pooling all parts of the IC in the density measurements to determine the percentage of VIP that are glutamatergic and the percentage of IC neurons that are VIP positive is not a good approach (subsection “VIP neurons are glutamatergic and represent 1.8% of IC neurons”). The distribution of the VIP cells across the IC should be mapped and reported, and examined in relation to the location of specific IC inputs. This should help to strengthen conclusions about the possible role of the VIP neurons in auditory processing. This is also a potential problem with the percentages of neurons with different intrinsic properties (subsection “VIP neurons exhibit sustained firing patterns and their intrinsic physiology varies along the tonotopic axis of the ICc”).

We agree that pooling VIP neurons from across IC subdivisions obscured underlying patterns in the VIP neuron distribution. To address this concern, we repeated the stereological analysis of VIP neuron density in IC sections. Data were prepared from two new VIP-IRES-Cre x Ai14 mice. Sections from these mice were stained with NeuN and anti-bNOS, the latter enabling us to differentiate the ICc from IC shell regions (Coote and Rees, 2008). The results of this experiment are shown in Table 2 and described in the Results, paragraphs twp and three of subsection “VIP neurons are glutamatergic and represent 3.5% of ICc neurons”. Note that previous results were discarded because tissue from that experiment was not stained for bNOS. However, the overall prevalence of VIP neurons in the IC was similar between the previous and new data sets (1.8 ± 0.3% in the previous data set versus 2.3 ± 0.3% in the new data set).

The results shown in Table 1 demonstrated that out of 793 tdTomato+ VIP neurons, only 10 (1.3%) stained with the antibody against GAD67. There was no apparent pattern to the distribution of the 10 tdTomato+ neurons that stained for GAD67, and we suspect these neurons were non-specifically labeled or represent rare instances where tdTomato was ectopically expressed (i.e. non-VIP neurons that had a low level of Cre expression at some point during development). Overall, we believe our data strongly support the conclusion that VIP neurons are glutamatergic across all subdivisions and regions of the IC.

We thank the reviewers for pointing out that the functional domain hypothesis might explain the distribution of VIP neurons in the ICc. This is a very important point and one that we believe significantly increases the impact of our study. We have added text to the Introduction (fourth paragraph), Results (paragraph three of subsection “VIP neurons are glutamatergic and represent 3.5% of ICc neurons”), and Discussion (paragraph one and subsection “The distribution and sources of input to VIP neurons suggest they specialize in monaural processing”) to indicate that the distribution of VIP neurons suggests they may belong to the set of monaural neurons that mainly receive input from the cochlear nucleus and VNLL and not from the superior olivary complex (Group 2 in Cant and Benson, 2006; Cluster 1 in Loftus et al., 2010).

3) Both the contralateral DCN and the contralateral IC are major sources of input to gerbil IC neurons (Nordeen et al., 1983; Moore et al., 1998), so it is perhaps not so surprising that this is true for mouse VIP neurons as well. Sustained firing is the most common firing pattern. The projections of the VIP neurons are investigated and documented in insufficient detail to tell whether these are special compared to other IC neurons. Overall, it seems that VIP neurons are an uncommon neuron type (<2%) with rather common properties, and, based on the present results, it is therefore not straightforward to infer their unique contribution to auditory processing by the IC. This needs to be addressed, including through the approaches suggested above.

We believe that addressing reviewer concerns has resulted in a manuscript that much better highlights the potential functional significance of VIP neurons. Here, in list form, we summarize several of the features of VIP neurons that are now directly addressed in the revised manuscript. We believe these features strongly suggest that VIP neurons play influential roles in auditory processing:

• By reassessing the density of VIP neurons with respect to IC subdivision boundaries, we found that VIP neurons represent 3.5% of ICc neurons. Since it is estimated that 15 – 20% of ICc neurons are stellate neurons, VIP neurons represent ~18 – 23% of ICc stellate neurons. Thus, VIP neurons encompass a significant percentage of the overall population of ICc stellate neurons. For reference, VIP neurons account for ~13% of inhibitory interneurons in cerebral cortex (Pronekke et al., 2015), but powerfully shape the function and output of cortical circuits (e.g. Pi et al., 2013; Garcia-Junco-Clemente et al., 2017, doi: 10.1038/nn.4483). We think the same is likely to hold true for VIP neurons in the IC.

• The asymmetrical distribution of VIP neurons in the ICc indicates that VIP neurons are present at highest densities in regions of the ICc that mainly receive monaural input, i.e. input from the DCN, AVCN, and VNLL, but not the LSO. This suggests that VIP neurons may play a specialized role in the processing of monaural stimuli. Furthermore, the Chen et al., 2018 study showed that excitatory ICc neurons that receive input from the DCN form a unique cluster of neurons that are relatively unlikely to receive input from other ascending auditory sources. We therefore propose that VIP neurons may play a critical role processing the output of the DCN.

• VIP receptors are present in the MGB and the SOC, both of which are postsynaptic targets of VIP neurons. Previous studies show that VIP receptors can powerfully influence the output of thalamocortical neurons. These results suggest that IC VIP neurons may exert outsize influence on their postsynaptic targets through activation of VIP signaling pathways.

• Almost nothing is known about the functional roles of ICc stellate neurons under in vivo conditions. This is largely because it has previously not been possible to target in vivo recordings to stellate neurons. The identification of VIP neurons in the IC and the existence of a mouse line that allows selective manipulation of VIP neurons (i.e. VIP-IRES-CRE mice) mean that our study positions the field for the first time to directly investigate the role of ICc stellate neurons in auditory processing.

4) The authors make the advantages of the optogenetic approach in Figure 7 over simple electrical stimulation insufficiently clear. As there is no selectivity in the labelling and little or no selectivity in the activation, what is the added value for activating the contralateral projections?

We suspect that electrical stimulation of commissural axons is just as likely to activate axons originating from the IC ipsilateral to the recording site as it is to activate the desired population of axons originating from the contralateral IC. Since most IC neurons have local axon collaterals, antidromic spikes in commissural axons could easily lead to synaptic events driven by ipsilateral neurons. In addition, the commissure contains axons that originate outside the IC; these axons could also be activated by electrical stimulation of the commissure. The optogenetic approach has a distinct advantage over electrical stimulation because we can verify that the viral transfection led to expression of Chronos only in neurons on one side of the IC (in our case the right IC, contralateral to the recording site in the left IC). Because of this, optical stimulation provides a cleaner data set for investigating commissural input because it prevents contamination from local, ipsilateral inputs or from extra-collicular inputs, which may have very different physiological properties. We added text detailing this rationale to the Results, subsection “VIP neurons in the ICc receive excitatory and inhibitory synaptic input from the contralateral IC”.

The optogenetic approach also has disadvantages, e.g. how meaningful is an EPSP rise time of 4.2 ms for a 2-5 ms pulse of light? It seems that a minimal stimulation protocol was used (subsection “Channelrhodopsin-assisted circuit mapping”), but without more details on the criteria to set duration and intensity, it is hard to assess whether the PSP kinetics are meaningful. As a minimum, more details are required to assure us that release was driven by action potentials (or if this is not possible do the TTX control).

We agree that optogenetic stimulation can distort the kinetics of PSPs. This is one of the reasons we work with Chronos, which has much faster activation kinetics than ChR2, and which therefore may increase the probability of Chronos eliciting action potential-driven synaptic release (Klapoetke et al., 2014). We now state this concern in the Results, paragraph six of subsection “VIP neurons in the ICc receive excitatory and inhibitory synaptic input from the contralateral IC”. We also performed additional experiments to address this concern. First, we directly compared the kinetics of light-evoked and electrically-evoked commissural PSPs, finding relatively small differences between the two. Second, we added a TTX control to the DCN CRACM experiment. We found that TTX completely abolished light-evoked EPSPs in 3 out of 3 VIP neurons (Results, paragraph four of subsection “VIP neurons in the ICc receive synaptic input from the DCN”). This suggests that optogenetic stimulation drove spike-mediated synaptic release, as opposed to directly depolarizing synaptic terminals. Third, we added additional details about the minimal stimulation paradigm to the Materials and methods, subsection “Channelrhodopsin-assisted circuit mapping”. Interestingly, although we always used the minimal light intensity and duration needed to elicit a PSP, we found that increasing light duration to 5 ms and using the maximal light intensity possible with our equipment (48 mW/mm2) did not significantly alter the kinetics of light-evoked EPSPs. Together, these results suggest that the PSP kinetics we observed with optogenetic stimulation are reasonable approximations of the PSP kinetics that exist under physiological conditions. Nonetheless, we hope that the improved clarity about the possible shortcomings of optogenetic stimulation make it easy for readers to make their own assessment of the data.